# Activity patterns of serotonin neurons underlying cognitive flexibility

**Sara Matias[1,2†], Eran Lottem[1†], Guillaume P Dugué[3], Zachary F Mainen[1*]**

[1]Champalimaud Research, Champalimaud Centre for the Unknown, Lisbon, Portugal; [2]MIT-Portugal Program, Porto Salvo, Portugal; [3]Institut de Biologie de l'Ecole Normale Supérieure, Centre National de la Recherche Scientifique, UMR8197, Institut National de la Santé et de la Recherche Médicale, Paris, France

**Abstract** Serotonin is implicated in mood and affective disorders. However, growing evidence suggests that a core endogenous role is to promote flexible adaptation to changes in the causal structure of the environment, through behavioral inhibition and enhanced plasticity. We used long-term photometric recordings in mice to study a population of dorsal raphe serotonin neurons, whose activity we could link to normal reversal learning using pharmacogenetics. We found that these neurons are activated by both positive and negative prediction errors, and thus report signals similar to those proposed to promote learning in conditions of uncertainty. Furthermore, by comparing the cue responses of serotonin and dopamine neurons, we found differences in learning rates that could explain the importance of serotonin in inhibiting perseverative responding. Our findings show how the activity patterns of serotonin neurons support a role in cognitive flexibility, and suggest a revised model of dopamine–serotonin opponency with potential clinical implications.

**\*For correspondence:** zmainen@ neuro.fchampalimaud.org

[†]These authors contributed equally to this work

**Competing interests:** The authors declare that no competing interests exist.

## Introduction

Serotonin (5-HT) is classically known to be implicated in mood and affective disorders (*Dayan and Huys, 2009*; *Cools et al., 2011*; *Li et al., 2012*), but it also plays a fundamental role when organisms need to adapt to sudden changes in the causal structure of an environment, such as during extinction and reversal learning paradigms (*Clarke et al., 2004, 2007*; *Boulougouris and Robbins, 2010*; *Bari et al., 2010*; *Brigman et al., 2010*; *Berg et al., 2014*). These studies have shown that 5-HT depletion, particularly in the orbitofrontal cortex (OFC) of primates, causes perseverative errors, that is, difficulties in stopping responses to previously rewarded stimuli which are no longer reinforced, without affecting learning of new associations or retention of learned associations (*Clarke et al., 2007*). Such results seem to stem from two functions of endogenous 5-HT activation: inhibiting learned responses that are not currently adaptive (*Soubrié, 1986*; *Bari and Robbins, 2013*) and driving plasticity to reconfigure them (*Maya Vetencourt et al., 2008*; *Jitsuki et al., 2011*; *He et al., 2015*). These mirror dual functions of dopamine (DA) in invigorating reward-related responses (*Niv et al., 2007*; *Panigrahi et al., 2015*) and promoting plasticity that reinforces new ones (*Tsai et al., 2009*; *Kim et al., 2012*; *Steinberg et al., 2013*). However, while DA neurons are known to be activated by reward prediction errors (*Schultz et al., 1997*; *Cohen et al., 2012*; *Eshel et al., 2015*), consistent with theories of reinforcement learning (*Sutton and Barto, 1998*; *Schultz et al., 1997*), the reported firing patterns of 5-HT neurons (*Liu et al., 2014*; *Cohen et al., 2015*; *Li et al., 2016*) do not accord with any existing theories (*Daw et al., 2002*; *Boureau and Dayan, 2011*; *Cools et al., 2011*; *Nakamura, 2013*). Indeed, 5-HT neurons have been proposed to signal worse-than-expected outcomes by being activated by negative reward prediction errors in the reinforcement learning framework (*Daw et al., 2002*; *Boureau and Dayan, 2011*), but there is little experimental evidence for such a signal in 5-HT neurons (*Cohen et al., 2015*; *Hayashi et al.,*

**eLife digest** Serotonin is a molecule that plays various roles in the human body. In the brain, it is involved in regulating mood and emotions. Growing evidence suggests that serotonin also helps animals – including humans – adapt their behavior to changes in their environment. To allow for such behavioral flexibility, serotonin might promote changes in the underlying brain structures and activity.

In a type of learning known as 'reversal learning', for instance, it is necessary to adapt to a sudden change in a previously familiar environment. For example, if there were a road closure on a person's way to work, they might want to learn to stop following their usual route and learn a new and better one. Previous research has shown that when serotonin signaling is reduced, people persevere. That is, they will keep following the old route even if it is no longer the best choice. How this process works is still largely unknown.

To start unraveling these mechanisms, Matias et al. trained mice in a reversal learning task while manipulating and recording the activity of the neurons that produce serotonin. The results showed that when the activity in serotonin neurons was experimentally blocked, the mice tended to keep looking for a reward that was no longer available. Then, by recording the activity of serotonin neurons, Matias et al. found that it was the surprise of discovering a change in a previously familiar environment that activates serotonin neurons. It did not matter whether the change was better or worse than expected. The findings suggest that together with dopamine, another molecule involved in learning from rewards, serotonin could play an important role during reversal learning.

One next step will be to determine if serotonin mainly stops perseverance in its tracks, or whether it works by helping to unlearn the old behavior, or a combination of both. In the future, this could further our understanding of depression, which can be viewed as a disorder characterized by patients being unable to adapt to adverse situations, leaving them trapped to repeat behaviors and thoughts that are not beneficial. Future studies could also build on these findings to guide the development of new treatments for depression that involve serotonin.

*2015*; *Li et al., 2016*) and 5-HT activation does not appear to drive aversive learning processes (*Dugué et al., 2014*; *Liu et al., 2014*; *McDevitt et al., 2014*; *Qi et al., 2014*; *Miyazaki et al., 2014*; *Fonseca et al., 2015*) the way DA drives appetitive learning (*Tsai et al., 2009*; *Kim et al., 2012*; *Steinberg et al., 2013*).

To investigate how 5-HT neurons could be involved in cognitive and behavioral flexibility in changing environments, we recorded their activity over several days in mice engaged in a reversal learning task in which the associations between neutral odor cues and different positive and negative outcomes are first well-learned and then suddenly changed. We reasoned that the scarcity of prediction error–like responses in previous recordings of identified 5-HT neurons (*Liu et al., 2014*; *Cohen et al., 2015*; *Li et al., 2016*) or unidentified raphe neurons (*Ranade and Mainen, 2009*; *Hayashi et al., 2015*) might be due to inadequately strong prediction errors. In these studies, the omission of rewards in a small fraction of trials was used to generate prediction errors. While increasing the variability of the outcome, this results in *expected uncertainty*. In contrast, in a reversal task, there is an abrupt violation of previously stable predictions and a step increase in the frequency of the prediction errors, termed *unexpected uncertainty*. Expected and unexpected uncertainty may differentially activate neuromodulatory systems (*Yu and Dayan, 2005*).

## Results

### Pharmacogenetic inactivation of DRN 5-HT neurons slows negative reversal learning

We first sought causal evidence that 5-HT neurons were linked to reversal learning in mice engaged in such a task by using a pharmacogenetic approach to silence 5-HT neurons (*Ray et al., 2011*; *Teissier et al., 2015*; *Armbruster et al., 2007*). Transgenic mice expressing CRE recombinase under the 5-HT transporter promoter (*Gong et al., 2007*) (SERT-Cre, n = 8) were transduced with a Cre-

dependent adeno-associated (AAV.Flex) virus expressing the synthetic receptor Di (DREADD, hM4D) (*Armbruster et al., 2007*) injected in the dorsal raphe nucleus (DRN), the major source of 5-HT to the forebrain (*Figure 1A*). These mice and their wild-type littermates (WT, n = 4) were trained in a head-fixed classical conditioning paradigm in which one of four odor cues (conditioned stimuli, CSs) was randomly presented in each trial. After a fixed 2 s trace period, each odor was followed by a tone and a specific outcome, or unconditioned stimulus (US) (*Figure 1B* top). For two odors the US was a water reward, and for the other two it was nothing (that is, only the tone was played). After training, mice showed learning of the odor–outcome contingencies, as indicated by differences in the anticipatory lick rate (*Figure 1B* bottom).

To test the impact of inhibiting DRN SERT-Cre expressing neurons (hereafter simply '5-HT neurons') we used a within-animal cross-over design in which each mouse experienced two reversals (*Figure 1C* top), receiving the DREADD ligand clozapine-*N*-oxide (CNO) during one and vehicle during the other; WT mice, which always received CNO, served as additional controls (*Figure 1—figure supplement 1A*). As expected, mice adjusted their anticipatory licking according to the new

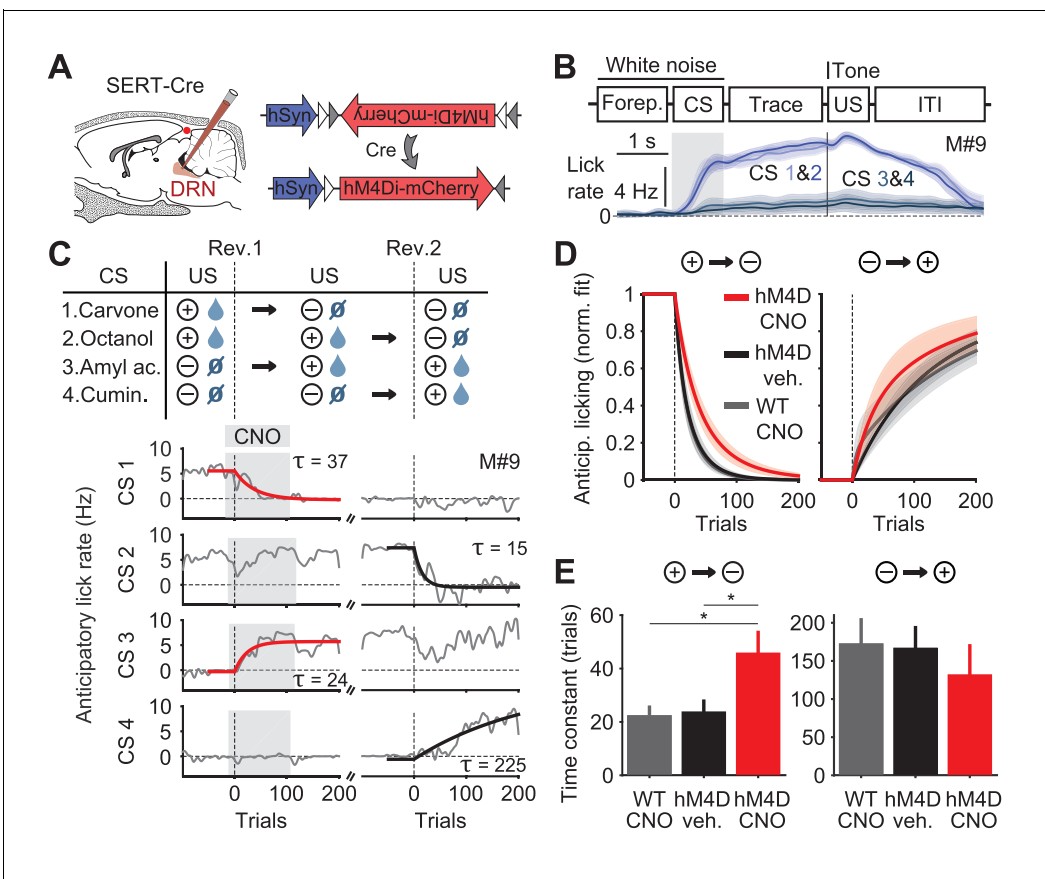

**Figure 1.** Inhibition of DRN 5-HT neurons causes perseverative responding. (A) Injections of Cre-dependent hM4Di-mCherry (right) in the dorsal raphe nucleus (DRN) of SERT-Cre mice (left). (B) Trial structure of the task (top) and mean lick rate of an example session along the four trial types (bottom). (C) Reversal procedure (top) and example of adaptation in mean anticipatory licking (baseline lick rate subtracted) across trials around reversals (bottom, gray), with exponential fits to the reversed odors (red and black traces). Gray shade represents the trials of sessions after CNO injection. (D) Mean exponential fits of anticipatory licking for each group of mice after reversal. (E) Mean time constants for the groups in (D) (one-way ANOVA, $F_{2,19} = 6.28$, p=0.008 for negative reversal, $F_{2,16} = 0.34$, p=0.715 for positive reversal; multiple comparisons indicated in the figure). *p<0.05.

The following figure supplement is available for figure 1:

**Figure supplement 1.** Anticipatory licking is more perseverative when DRN 5-HT neurons are inhibited.

associations in both reversals (*Figure 1C* bottom, gray traces). For worse-than-expected outcomes (negative reversals), the kinetics of adaptation to the new contingencies were significantly slower in hM4D mice receiving CNO, compared to hM4D no-CNO controls and WT controls (*Figure 1C,D,E*; *Figure 1—figure supplement 1B,C*). In contrast, for better-than-expected outcomes (positive reversals), there was no significant difference between treatment and control groups (*Figure 1D,E*).

This experiment shows that a population of 5-HT neurons in the DRN contributes to inhibiting perseverative responding, suggesting an anatomical and genetic substrate for previous results obtained with pharmacological and lesion experiments (*Clarke et al., 2004*, *2007*; *Boulougouris and Robbins, 2010*; *Bari et al., 2010*; *Brigman et al., 2010*). These findings also defined an access point to assess how the net activity of a specific population of 5-HT neurons could account for its effects on reversal learning.

## Photometric monitoring of DRN 5-HT activity patterns in a reversal task

To obtain a broad view of DRN 5-HT activity and compare our results to other DRN recording studies (*Hayashi et al., 2015*; *Cohen et al., 2015*), for the next series of recording experiments we used a second reversal task in which mice learned to associate four odors with four different outcomes: a

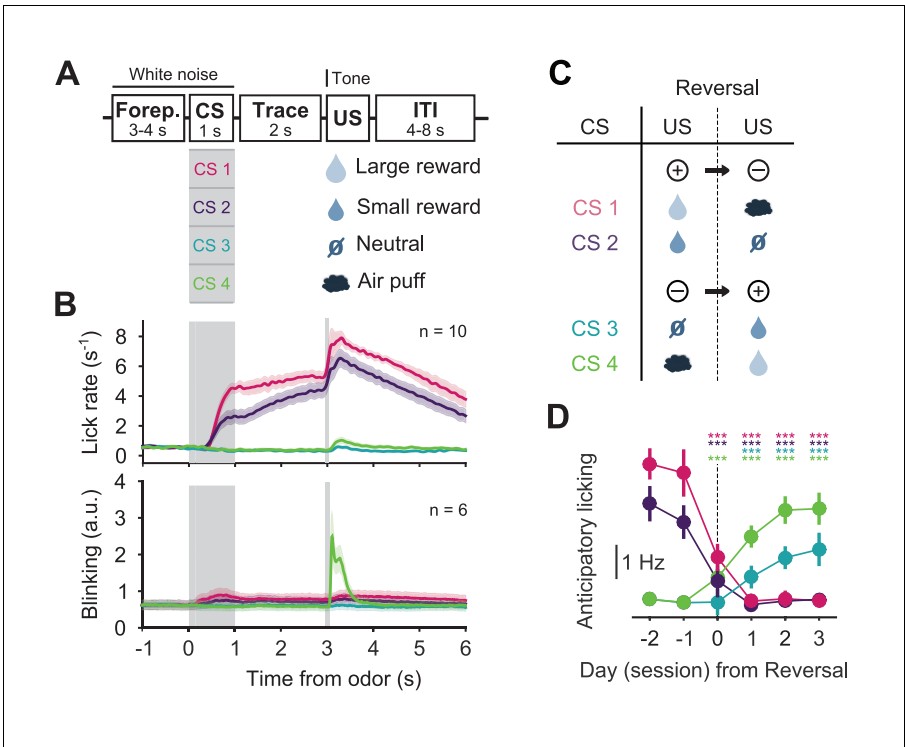

**Figure 2.** Behavior of head-fixed mice trained in a reversal task. (**A**) Schematics of the trial structure in the classical conditioning task (before reversal) with four different outcomes. In each trial, one of four odors was randomly selected and presented for 1 s after a variable foreperiod (Forep). The associated outcome was delivered after a 2 s trace period, together with a tone (same tone for all trial types). Mice were presented with 140 to 346 interleaved trials (mean ± SD: 223 ± 30) per session (day). (**B**) Top: Mean lick rate of SERT-Cre mice in this task (n = 10) along the duration of each trial type. For each mouse, three sessions of the classical conditioning task where initial associations had already been learned were averaged. Bottom: Mean eye movement of SERT-Cre mice (n = 6) along the duration of each trial type. Shaded areas represent s.e.m. (**C**) Reversal of CS–US contingencies (negative reversal: CS 1 and 2; positive reversal: CS 3 and 4). (**D**) Anticipatory licking (mean of 500–2800 ms after odor onset, after subtracting the baseline) across mice for sessions around reversal, showing that the lick rate triggered by the presentation of each odor is adjusted after reversal (n = 8, two-way ANOVA with factors day (days −2 and −1 are considered together) and mouse, main effect of day: $F_{4,2597}$ = 722.14, p<0.001 for odor 1, $F_{4,2554}$ = 355.53, p<0.001 for odor 2, $F_{4,2513}$ = 104.93, p<0.001 for odor 3, $F_{4,2559}$ = 381.55, p<0.001 for odor 4). Colors follow odor identity as in (**A**). ***p<0.001.

large water reward, a small water reward, nothing (neutral) and a mild air puff to the eye (*Figure 2A*). After approximately two weeks of training, mice showed robust CS-triggered anticipatory licking correlated to the reward value of the associated USs (large water > small water > neutral ≈ air puff) and eye-blink responses to the delivery of air puffs (*Figure 2B*). We then reversed the CS–US associations in pairs, such that the CSs associated with the large and small rewards now predicted the air puff and neutral outcomes, respectively, and vice versa (*Figure 2C*). Upon this reversal, mice experienced strong violations of CS-based expectations (unexpected uncertainty), both positive and negative in value, when the unexpected USs were delivered. Anticipatory licking measurements showed that mice adapted to reversal of contingencies over 1–3 additional sessions (*Figure 2D*).

To record the population activity of 5-HT neurons across days around the time of the reversal, we used photometry to monitor the activity of these DRN 5-HT neurons through an implanted optical fiber (*Tecuapetla et al., 2014*) (*Figure 3A*). SERT-Cre mice were infected in the DRN using two AAV.Flex viruses containing the genetically-encoded calcium indicator GCaMP6s (*Chen et al., 2013*) and the activity-insensitive fluorophore, tdTomato (*Figure 3B,C*). We verified the specificity of GCaMP6s expression to DRN 5-HT neurons using histological methods (*Figure 3—figure supplement 1*). We used a regression-based method to decompose the dual fluorescence signals into a GCaMP6s-specific component, reflecting activity-dependent changes, and a shared component, reflecting general fluorescence changes (for example, movement artifacts; see Methods and *Figure 3—figure supplement 2*). We validated the effectiveness of this approach in control mice (n = 4) infected in the DRN with yellow fluorescent protein (YFP; replacing GCaMP) and tdTomato (*Figure 3—figure supplements 1* and *2*).

Before reversal, photometric 5-HT responses were similar to previous electrical (*Liu et al., 2014*; *Cohen et al., 2015*) and photometric (*Li et al., 2016*) recordings of identified 5-HT neurons: 5-HT neurons were activated by reward-predicting CSs and air puffs (*Figure 3D*, *Figure 3—figure supplement 3*). YFP control mice implanted and recorded in the same manner showed no photometric responses to these events (*Figure 3—figure supplement 4*). To compare directly how DA neurons respond in the same paradigm, we infected TH-Cre mice and targeted neurons in either the posterior lateral ventral tegmental nucleus (VTA) or the substantia nigra pars compacta (SNc) (*Figure 3E*, *Figure 3—figure supplement 5*). DA photometry responses in these two areas were similar and were therefore combined. As expected, DA neurons were activated by reward-predicting cues, and showed small responses to predicted rewards (*Figure 3F*).

## DRN 5-HT neurons respond to both positive and negative US prediction errors

To understand the pattern of 5-HT neural activity that could underlie adaptation to reversal of contingencies, we first analyzed US responses, which could contribute to or modulate reinforcement learning. In general, we found that the abrupt reversal of cue–outcome associations caused immediate changes in 5-HT and DA US responses, much more so than in reward omission tests (*Ranade and Mainen, 2009*; *Cohen et al., 2015*; *Hayashi et al., 2015*; *Li et al., 2016*), consistent with sensitivity to the sudden increase in uncertainty that occurred upon reversal after extensive training.

We first examined the case of positive reversals. 5-HT neurons showed little or no response to large water rewards before reversal when they were predicted by the preceding CS, but responded robustly to the same events when they were unpredicted, after reversal (*Figure 4A,B*). Thus, 5-HT neurons showed an excitatory response to a better-than-expected outcome, or positive reward prediction error (RPE). The response to the small reward was also modulated by reward expectation (*Figure 4—figure supplement 1*), although to a lesser degree, perhaps due to the presence of a small response even after extensive training (*Figure 3—figure supplement 3B*). Like 5-HT neurons, DA neurons also showed stronger excitatory responses to water rewards immediately after reversal when they violated cue-based predictions, as opposed to before reversal when they occurred as predicted (*Figure 4C*, *Figure 4—figure supplement 1C*). Therefore, both 5-HT and DA neurons showed an increase in activity in response to positive RPEs, and both showed a larger response for the larger magnitude RPE.

We next examined the response to the neutral USs. Before reversal, this US elicited little response from either 5-HT or DA neurons. After reversal, the neutral US was presented when a small water

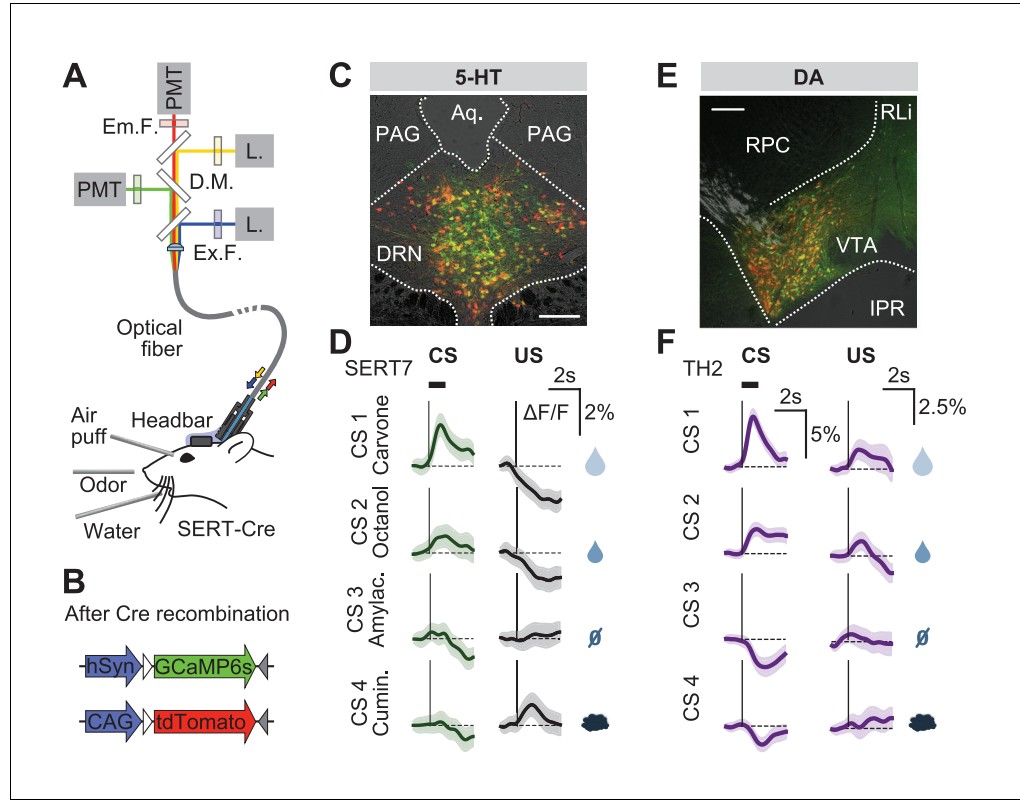

**Figure 3.** Responses of 5-HT and DA neurons before reversal. (**A**) Fiber photometry with movement artifact correction in head-fixed mice. L: laser; PMT: photomultiplier tube; D.M: dichroic mirror; Ex: excitation; Em: emission; F: filter. (**B**) Cre-dependent fluorophores used. (**C**) Coronal section showing expression of GCaMP6s and tdTomato in the DRN of a SERT-Cre mouse (scale bar: 200 μm). PAG: periaqueductal gray; Aq: Aqueduct. (**D**) Mean responses of 5-HT neurons to the four CSs and USs during an example session of a mouse before reversal. Shaded areas represent 95% confidence interval (CI). (**E**) Coronal section showing expression of GCaMP6s and tdTomato in the ventral tegmental area (VTA) of a TH-Cre mouse (scale bar: 200 μm). RLi: rostral linear nucleus of the raphe; RPC: red nucleus, parvicellular part; IPR: interpeduncular nucleus. (**F**) Mean responses of DA neurons to the four CSs and USs during an example session of a mouse before reversal. Shaded areas represent 95% CI.

The following figure supplements are available for figure 3:

**Figure supplement 1.** Expression of GCaMP6s and of tdTomato in DRN 5-HT neurons.

**Figure supplement 2.** Linear regression approach to eliminate movement artifacts from neuronal photometric data.

**Figure supplement 3.** Responses of DRN 5-HT neurons to odor cues and to predicted outcomes.

**Figure supplement 4.** Fluorescence changes to odor cues and to predicted outcomes in YFP control mice.

**Figure supplement 5.** Responses of midbrain DA neurons before reversal.

reward was predicted. Therefore, it represented a reward omission or negative RPE. Interestingly, 5-HT neurons showed a robust excitatory response to the neutral US after reversal (*Figure 5B*). In contrast, DA neurons showed an inhibitory response to the same event (*Figure 5C*).

Taking the neutral and rewarding USs together, the results show that midbrain DA neurons respond to positive and negative RPEs with modulation of the opposite sign, as reported previously in reward omission paradigms (*Cohen et al., 2012*; *Schultz et al., 1997*); but see *Matsumoto and Hikosaka (2009)*; *Lammel et al. (2011)*; *Kim et al. (2016)*; *Matsumoto et al., 2016*). On the other

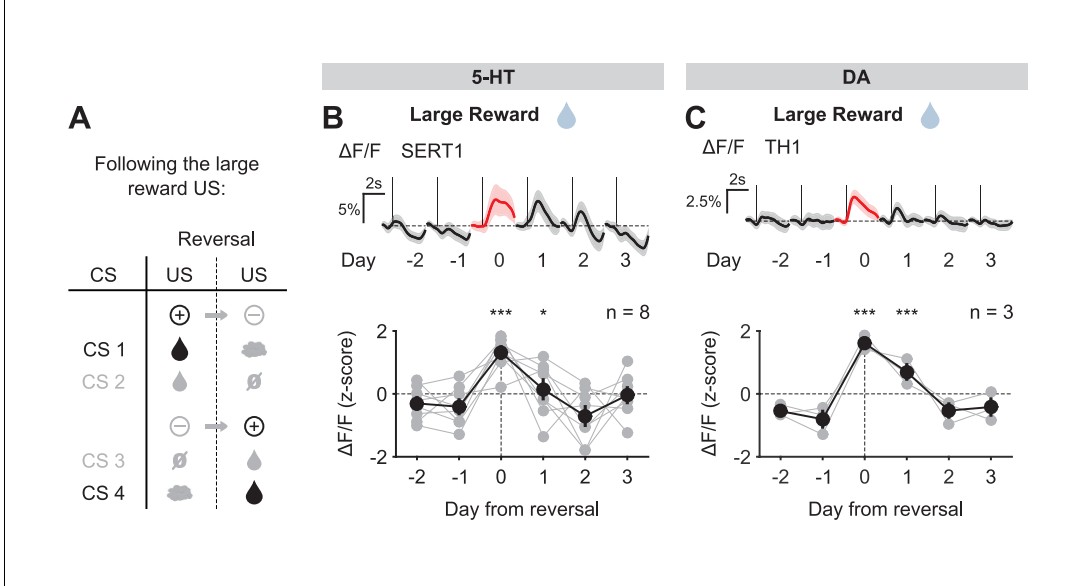

**Figure 4.** US responses of 5-HT and DA neurons to the large reward during reversal. (**A**) Schematic of the reversal procedure following the large reward US. (**B**) Top: Mean large reward US responses of an example mouse (SERT1) across days around reversal (shaded areas represent 95% CI); Bottom: change in mean large reward response amplitude (z-scored across days): gray dots represent individual mice (n = 8), black dots average (± s.e.m.) of mice (two-way ANOVA with factors day and mouse, the main effect of day is $F_{4,2592} = 31.47$ p<0.001; multiple comparisons with the two days before reversal, corrected using Scheffé's method, are indicated in the figure). (**C**) Same as (**B**) for DA neurons (n = 3 mice): $F_{4,853} = 32.46$, p<0.001. *p<0.05, ***p<0.001.

The following figure supplement is available for figure 4:

**Figure supplement 1.** US responses of 5-HT and DA neurons to small reward during reversal.

hand, SERT-positive DRN 5-HT neurons show excitatory responses to both positive and negative RPEs. Thus, DRN 5-HT responses to rewards and reward omissions resemble an 'unsigned RPE' or 'surprise' signal (see Discussion).

Finally, we examined the response of 5-HT and DA neurons to predicted and unpredicted air puffs. In contrast to other USs, DRN 5-HT neurons were mildly activated by air puff USs, even after extensive training (*Figure 3*; *Figure 3—figure supplement 3B*). Upon reversal, despite the fact that the air puff US now represented a large negative RPE (since the large water reward was predicted), 5-HT neurons showed no significant response (*Figure 5—figure supplement 1B*). Midbrain DA neurons, on the other hand, showed no response to the air puff US after training, but showed a small but significant inhibitory response after reversal (*Figure 5—figure supplement 1C*).

The results for all USs are summarized in *Figure 6*. Overall, midbrain DA responses adhered closely to the model of a 'signed RPE', including for the air puff, whereas the DRN 5-HT neurons resembled an 'unsigned RPE' with respect to rewards and reward omissions, but they diverged from this model for air puff responses (see Discussion for further interpretation). Thus, 5-HT and DA neurons are both sensitive to violations of expectation that occur during an abrupt reversal, with the two systems responding in the same way to better-than-expected outcomes but in opposite ways to worse-than-expected outcomes.

## DRN 5-HT neurons are activated by out-of-context delivery of USs

To further investigate the idea that 5-HT neurons might report prediction errors, we examined responses to USs delivered outside of the normal context. For this, five days after reversal, on a small fraction (20%) of trials, a randomly-selected US was delivered at the time that a CS was normally presented (*Figure 7A*). We found that water rewards produced larger 5-HT responses when they were presented in this way, compared to when preceded by a well-learned cue (*Figure 7B*). Of particular interest was that even neutral tones produced an excitatory response when an odor was

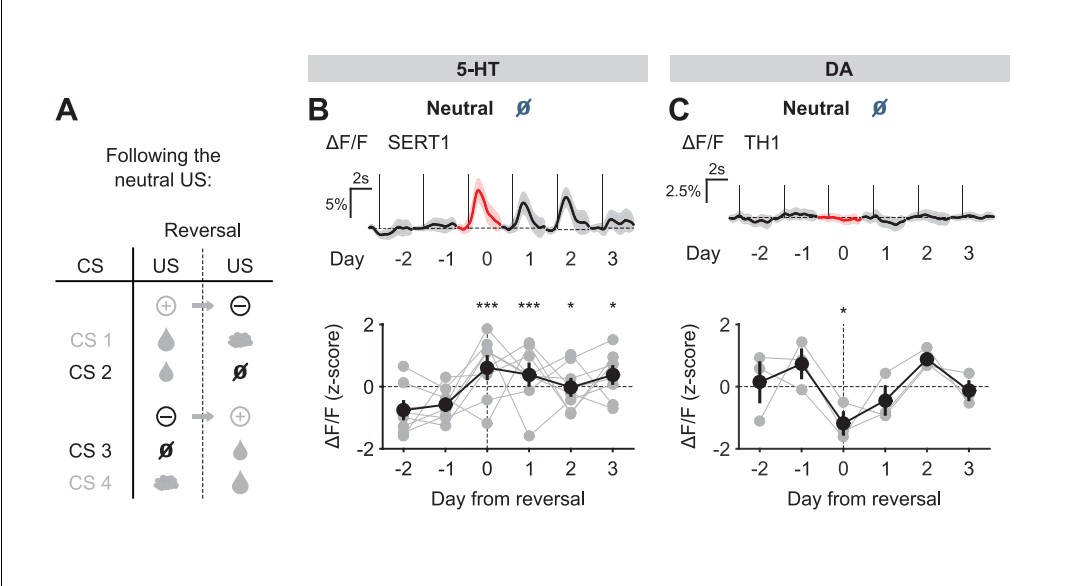

**Figure 5.** US responses of 5-HT and DA neurons to neutral outcome during reversal. (**A**) Schematic of the reversal procedure following neutral US. (**B**) Top: Mean neutral US responses of an example mouse (SERT1) across days around reversal (shaded areas represent 95% CI); Bottom: change in mean neutral response amplitude (z-scored across days): gray dots represent individual mice (n = 8), black dots average (± s.e.m.) of mice (two-way ANOVA with factors day and mouse, the main effect of day $F_{4,2535}$ = 10.71, p<0.001; multiple comparisons with the two days before reversal, corrected using Scheffé's method, are indicated in the figure). (**C**) Same as (**B**) for DA neurons (n = 3 mice): $F_{4,843}$ = 4.54, p=0.001. *p<0.05, ***p<0.001.

The following figure supplement is available for figure 5:

**Figure supplement 1.** US responses of 5-HT and DA neurons to air puff during reversal.

expected (*Figure 7B*; *Figure 7—figure supplement 1*). Therefore, 5-HT neurons were activated by the substitution of one neutral stimulus with another. DA neurons also responded strongly to uncued rewards, as previously reported (*Schultz et al., 1997*; *Cohen et al., 2012*), but little to other uncued

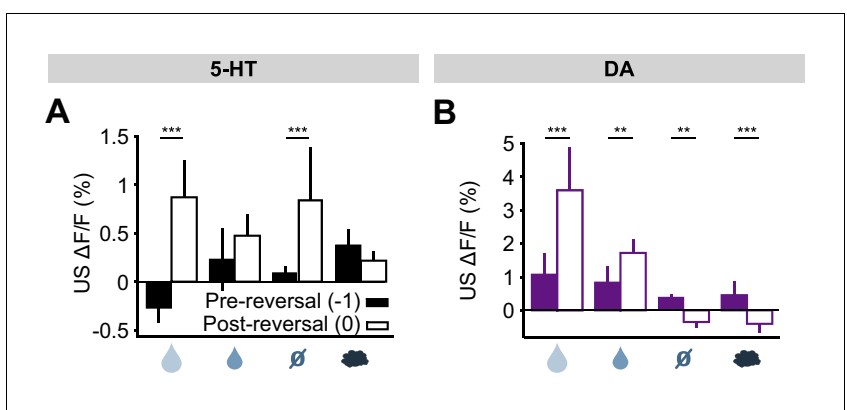

**Figure 6.** Responses of 5-HT and DA neurons to outcomes are differentially modulated by expectations. (**A**) Mean (± s.e.m.) response of 5-HT neurons, across mice, to the four USs before (day −1, filled bars) and right after (day 0, open bars) reversal (n = 8 mice, two-way ANOVA with factors mouse and day, the main effect of day $F_{1,764}$ = 84.36, p<0.001 for large reward, $F_{1,748}$ = 3.49, p=0.062 for small reward, $F_{1,756}$ = 38.17, p<0.001 for neutral, $F_{1,766}$ = 2.79, p=0.095 for air puff). (**B**) Same as (**A**) for midbrain DA neurons (n = 3 mice, $F_{1,249}$ = 67.9, p<0.001 for large reward, $F_{1,277}$ = 8.49, p=0.004 for small reward, $F_{1,278}$ = 10.95, p=0.001 for neutral, $F_{1,250}$ = 12.74, p<0.001 for air puff). **p<0.01, ***p<0.001.

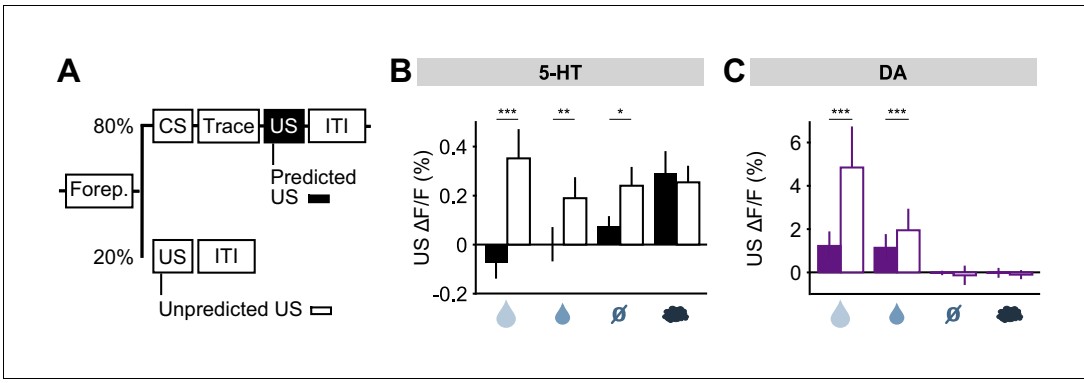

**Figure 7.** DRN 5-HT neurons respond more to uncued outcomes. (A) Behavioral task diagram. (B) Mean (± s.e.m.) response of DRN 5-HT neurons across mice to the four USs when they are predicted (filled bars) and when they are unpredicted (open bars) (n = 4 mice, two-way ANOVA with factors type (predicted or unpredicted) and mouse, the main effect of type: large reward $F_{1,923}$ = 45.17, p<0.001, small reward $F_{1,944}$ = 8.42, p=0.0038, neutral $F_{1,924}$ = 5.36, p=0.0208, air puff $F_{1,924}$ = 0.61, p=0.4331). (C) Same as (B) but for midbrain DA neurons (n = 3 mice, large reward $F_{1,642}$ = 175.05, p<0.001, small reward $F_{1,589}$ = 17.53, p<0.001, neutral $F_{1,673}$ = 0.52, p=0.4707, air puff $F_{1,601}$ = 0.34, p=0.5598). *p<0.05, **p<0.01, ***p<0.001.

The following figure supplement is available for figure 7:

**Figure supplement 1.** Responses of 5-HT and DA neurons to predicted and unpredicted outcomes.

USs (*Figure 7C*) (*Matsumoto et al., 2016*). Thus, consistent with the responses following CS–US reversal, this experiment also showed that, with respect to water rewards and reward omissions, 5-HT neurons respond in the same manner to unexpected events, whether negative, neutral or positive, whereas DA neurons are primarily sensitive to unexpected events that have some reward value.

## CS responses of 5-HT neurons have slower kinetics after reversal than DA neurons'

US responses are appropriate to drive learning across trials, but occur too late within a given trial to inhibit CS-driven behavioral responses directly. If it is to intervene in time to prevent a response, behavioral inhibition should be triggered by predictive CS cues. We therefore examined the CS responses of 5-HT and DA neurons carefully, to test how they might contribute to reversal learning.

Before reversal, both 5-HT and DA neurons showed CS responses that correlated with the relative value of the US predicted by the CS (large reward > small reward > neutral ≈ air puff) (*Figure 3*, *Figure 3—figure supplements 3* and *5*). After the reversal, both adjusted to the new contingencies such that, by three days post-reversal, the CS responses reflected their new US associations (*Figure 8*). Thus, despite small differences in their relative magnitudes, and in contrast to their distinct US responses, DA and 5-HT neurons showed CS responses that were remarkably similar, both before and after reversal learning. If DA and 5-HT have opposing direct effects on behavior (for example, *Cools et al., 2011*), these results suggest that they would simply cancel one another out.

However, when we analyzed the time course of the adaptation of the CS responses, we found that 5-HT CS responses had a markedly slower rate of adaptation to the new contingencies than did DA CS responses (*Figure 9A,B*, *Figure 9—figure supplements 1* and *2*). The difference in the time constant of CS adaptation was significant for both negative and positive reversals, and was not due to differences in learning rates between groups of mice (*Figure 9C*). We also tested whether US responses, which presumably reflect, in part, CS-related learning, also show a difference in the time course of adaptation. However, because the US signals showed a smaller signal-to-noise ratio than the CS signals, reliable time courses could not be extracted.

A potentially important consequence of the difference in CS learning time constants is that it implies an asymmetry between DA and 5-HT systems in positive and negative reversals (*Solomon and Corbit, 1974*). During a positive reversal, because the adaptation of 5-HT cue responses is much slower than that of DA cue responses, the net signal will be transiently biased

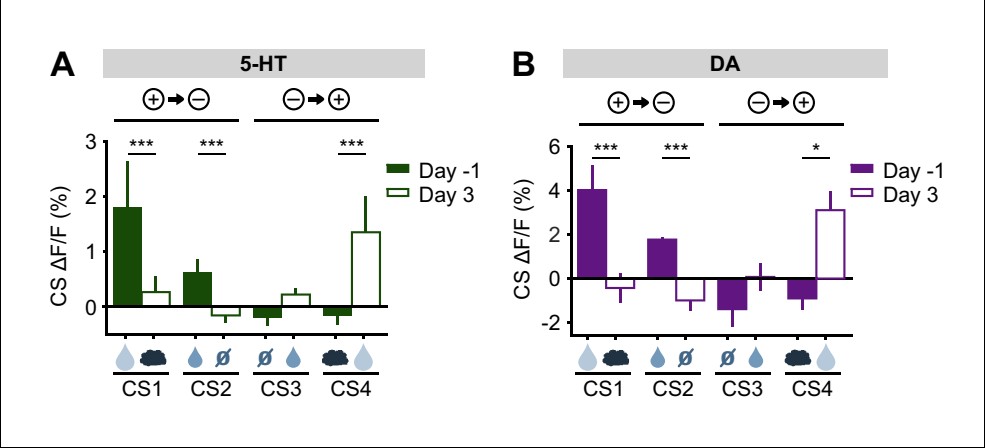

**Figure 8.** 5-HT and DA CS responses are relearned after the reversal. (**A**) Mean (± s.e.m.) response of 5-HT neurons across mice to the four CSs before reversal (filled bars) and after adaptation to the reversed contingencies (open bars) (n = 8 mice, two-way ANOVA with factors day and mouse, the main effect of day: large reward $F_{1,906}$ = 17.35, p<0.001, small reward $F_{1,902}$ = 14.87, p<0.001, neutral $F_{1,882}$ = 0.13, p=0.72, air puff $F_{1,914}$ = 17.12, p<0.001). (**B**) Same as (**A**) for midbrain DA neurons (n = 3 mice, large reward $F_{1,294}$ = 15.35, p<0.001, small reward $F_{1,336}$ = 71.72, p<0.001, neutral $F_{1,282}$ = 3.45, p=0.06, air puff $F_{1,312}$ = 6.56, p=0.01). *p<0.05, ***p<0.001.

towards the effects of DA (*Figure 9D*, right). Conversely, during a negative reversal, because 5-HT cue responses persist longer than those of DA, the difference will be biased towards the effects of 5-HT (*Figure 9D*, left). This suggests a novel mechanism by which 5-HT can contribute to preventing perseverative responding during negative reversals (*Clarke et al., 2007*), by directly inhibiting behavioral responses to CSs that have undergone decreases in associated outcome values.

The DREADD inactivation experiment (*Figure 1*) supported the contribution of 5-HT to negative reversal learning, but did not distinguish whether the relevant activity occurs during the CS or the US. To test for a contribution of the CS-related activity, we asked whether there was a correlation in the animal-to-animal variability in the time constant of behavioral adaptation (anticipatory licking) and neural adaptation (CS magnitude). Remarkably, we observed a significant correlation between the time constant of DRN 5-HT CS response and the time constant of CS-related licking for the negative reversals but not the positive one (*Figure 10*), suggesting that these responses could be involved in adapting to negative reversals. Moreover, during such negative reversals the time constant of DRN 5-HT responses was slower than that of anticipatory licking for all animals (*Figure 10*; see *Figure 10—figure supplement 1* for DA). We note that, while we expected that the adaptation of 5-HT CS responses to the reversal should be at least as slow as that of anticipatory licking for the two to be causally related, the fact that it was much slower (around eight times as slow) requires an explanation. One possibility is that our behavioral readout (that is, tongue protrusions long enough to be detected by our sensor) is just a 'tip-of-the-iceberg' of motor responses to appetitive cues, and that other, covert, movements also need to be suppressed by 5-HT during relearning, and thus 5-HT neurons need to be active until all motor responses to appetitive cues have disappeared. Alternatively, it may be the case that 5-HT CS responses could serve more than a mere motor suppression function during reversal learning, and contribute to the longer-lasting learning processes required for reversal learning (*He et al., 2015*), such as those that prevent spontaneous recovery following extinction training (*Karpova et al., 2011*).

## Discussion

We used a reversal learning task in head-fixed mice to study the role of 5-HT in adapting to the reversal of cue–outcome contingencies, a model of the cognitive flexibility required to adapt to dynamic environmental conditions. Pharmacogenetic inhibition of DRN 5-HT neurons showed that 5-HT activity contributes to preventing perseverative responses to formerly reward-predictive cues,

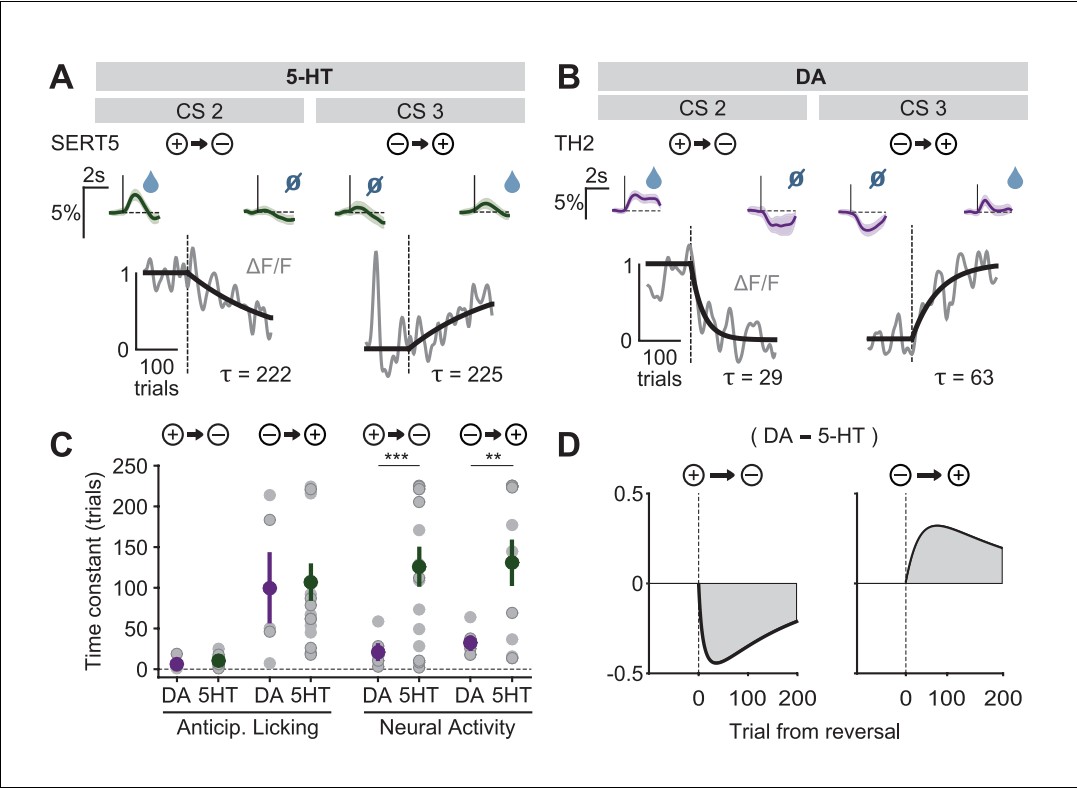

**Figure 9.** Distinct speed of CS reversal learning in DRN 5-HT and midbrain DA neurons. (**A**) Normalized exponential fits (black traces) to the mean amplitude of the CS responses (gray traces) across trials for CS 2 and CS 3 of an example SERT-Cre mouse. Insets on top show mean CS response (and 95% CI) on days −1 (left) and 3 (right). (**B**) Same as (**A**) for an example TH-Cre mouse. (**C**) Mean time constants (± s.e.m., green and purple dots) of the exponential fits of CS responses obtained for TH-Cre and SERT-Cre mice during reversal learning (neural activity: unpaired *t*-tests, p<0.001 for negative reversal, p=0.0023 for positive reversal; no significance obtained for anticipatory licking). Gray dots represent individual mouse–odor pairs for each category of reversal type; gray dots with darker edges represent odors 2 or 4, while the remaining dots represent odors 1 or 3. (**D**) Difference in the mean fitted amplitude of CS response between DA and 5-HT during negative reversal (left) and during positive reversal (right). **p<0.01, ***p<0.001.

The following figure supplements are available for figure 9:

**Figure supplement 1.** CS responses of DRN 5-HT neurons during reversal.

**Figure supplement 2.** CS responses of midbrain DA neurons during reversal.

consistent with previous work in rodents and primates (*Clarke et al., 2004, 2007*; *Boulougouris and Robbins, 2010*; *Bari et al., 2010*; *Brigman et al., 2010*; *Berg et al., 2014*; *Bari and Robbins, 2013*). These observations suggest two possible complementary contributions of 5-HT to behavioral flexibility: (1) to facilitate the learning of new associations and (2) to directly inhibit responses which are no longer appropriate. To elucidate how the dynamics of endogenous neural activity could support these functions, we used fiber photometry to monitor 5-HT and DA during reversal learning. This revealed two important findings.

## DRN 5-HT neurons are activated by both positive and negative reward prediction errors

First, we found that 5-HT US responses were strongly sensitive to changes in cue–outcome contingency after the reversal. Remarkably, 5-HT neurons responded with a similar transient excitation to violations of expectation that were either better-than-expected or worse-than-expected reward

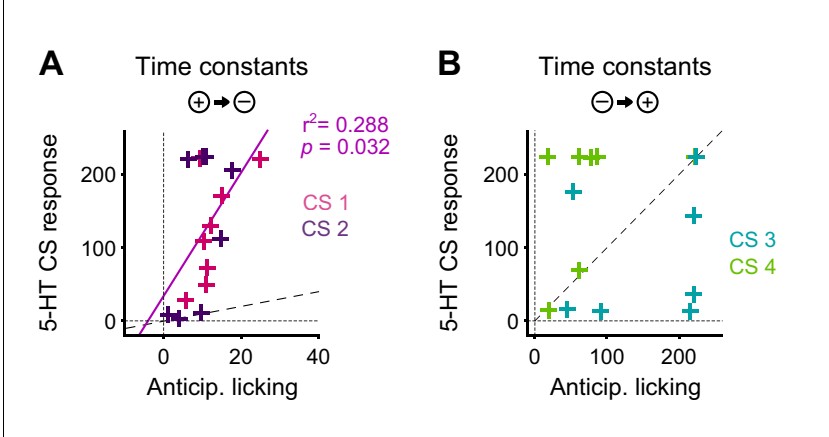

**Figure 10.** The correlation between the speed of DRN 5-HT cue learning and anticipatory licking. (**A**) Correlation between time constants of 5-HT CS responses and anticipatory licking for the negative reversal. A significant linear relationship was found: $y = 8.4*x + 34$; $r^2$: 0.288; F = 5.67, p=0.032. (**B**) Same as (**A**) for positive reversals (no relationship was found). Diagonal dashed lines represent $y = x$.

The following figure supplement is available for figure 10:

**Figure supplement 1.** Time constant of DA CS response versus time constant of corresponding anticipatory licking.

outcomes. Midbrain DA neurons, on the other hand, responded oppositely to better-than-expected and worse-than-expected outcomes. Thus, whereas DA neurons could be described as reporting a signed RPE, 5-HT neurons appeared to report, in part, an unsigned RPE (but see below for discussion of responses to aversive events). That is, 5-HT neurons were sensitive not to the direction of error but to its magnitude. These responses could also be described as a type of 'surprise' signal (for example, *Courville et al., 2006*). Supporting this idea, we found that 5-HT neurons were also sensitive to substitution of one neutral cue for a cue of another modality (sound for odor). It remains to be determined whether these responses were dictated entirely by small differences in reward value, or whether they reflect sensory as well as value prediction errors.

Unsigned prediction error signals have been proposed on theoretical grounds to be ideal for regulating learning and attention based on uncertainty (*Pearce and Hall, 1980*; *Courville et al., 2006*). By reporting such signals, 5-HT US responses would be suitable to drive plasticity and re-learning during reversal of contingencies. The strong excitatory response of the 5-HT system to negative RPEs, caused by reward omissions, provides a possible explanation for why inhibiting this system impairs negative reversal learning (*Clarke et al., 2007*; *Bari et al., 2010*) (*Figure 1*). That is, during negative reversals or extinction learning, the 5-HT system, either directly or through an interaction with the DA system (*Boureau and Dayan, 2011*), could facilitate trial-by-trial undoing of DA-dependent learning. Since 5-HT neurons also respond during positive prediction errors, such as during positive reversal or initial learning, such activation might compete with co-occurring DA signals, slowing positive learning, as has been described (*Fletcher et al., 1999*). The preferential involvement of 5-HT in 'unlearning' responses could be explained by the relative effects of 5-HT release on downstream targets, where 5-HT may favor long-term depression (LTD) and DA long term potentiation (LTP) (*He et al., 2015*).

## Response to aversive events by DRN 5-HT neurons

5-HT US responses contained one notable divergence from an idealized prediction error: air puff USs continued to evoke responses, even after extensive training, and showed only minor sensitivity to the presence of a predictive cue — observations consistent with a previous report (*Cohen et al., 2015*). One possible explanation is that mice failed to learn the predictive relationship between the CS and the air puff. Indeed, mice showed air puff–triggered blink responses, but failed to learn

*anticipatory* blinking responses despite extensive training. This result likely depends on the relatively long duration of the CS–US trace period, here 2 s (*Reynolds, 1945*; *Boneau, 1958*; *Cohen et al., 2012*, *2015*; *Caro-Martín et al., 2015*; cf. *Matsumoto et al., 2016*). This is consistent with the idea that mice did not learn the CS–air puff association. If mice formed no CS-dependent predictions about the air puff, then they might have experienced each air puff as 'unpredicted', whether before or after reversal. In this case, the presence of robust air puff US responses would be consistent with an unsigned value prediction error. However, since we have no explanation for how mice could succeed in learning a CS–reward association while failing to learn the CS–air puff association, other explanations should also be considered.

A second possible line of explanation for the observation that the air puff did not elicit an increased response after the reversal is that 5-HT neurons report at least two qualitatively distinct signals: one relating to the processing of rewards and the other to the processing of aversive stimuli. In principle, following a reversal from large reward to air puff, one would have expected a contribution of the reward omission response to the US response, as seen in the small reward to neutral reversal. The lack of such a response could indicate either simple saturation or a suppressive influence of the air puff on the reward omission signal. A distinction between the encoding of rewarding vs. aversive events by the DA system has been proposed (*Fiorillo, 2013*). The presence of dual signals might reflect the inclusion of multiple 5-HT neuronal populations within our photometric recordings. In future experiments, these could be distinguished using a pathway-specific labeling, as has been done in the DA system (*Lerner et al., 2015*; see further discussion below). On the other hand, VTA DA neurons have been reported to integrate reward and aversive outcome values, but with aversive responses being strongly modulated by the rate of reward available in the current context (*Matsumoto et al., 2016*). In future experiments, it will be important to understand how individual 5-HT neurons integrate information from combinations of outcomes, and in different reward contexts.

An alternative possibility is that the pattern of 5-HT US responses could be understood together as a variation on a prediction error signal. Whereas mice can control the consumption of available water, they cannot control the delivery of air puffs; they are afforded no means to escape in the head-fixed configuration. It is therefore interesting to consider the possibility that 5-HT neurons might report errors of *control* rather than errors of prediction. Under this hypothesis, an aversive outcome such as the air puff continues to generate a response in 5-HT neurons because the organism has not managed to control this aspect of its environment. If the mouse were offered a means to escape, we would expect to see the air puff response diminish. Conversely, because the 5-HT US response is also sensitive to errors of a positive nature, we would also expect to see continued responses to a non-controllable reward, for example, direct oral infusion of sucrose (*Li et al., 2016*). Such 'unsigned control errors' could provide the organism with a signal of the magnitude of cognitive or behavioral effort required to adapt to a given situation, a signal that could be read out for the purpose of energizing or deenergizing actions.

Consistent with the control error hypothesis, predictable but uncontrollable shocks robustly activate the immediate early gene *Fos* in DRN 5-HT neurons (*Takase et al., 2004*), and this activation is lowered by controllability signals from the ventral medial prefrontal cortex (*Bland et al., 2003*; *Amat et al., 2005*). This proposal also finds support in a recent study showing that DRN 5-HT activity mediates short-term sensorimotor adaptation in zebrafish, by reporting the difference between the expected and actual sensory consequences of motor commands (*Kawashima et al., 2016*). However, further experiments will clearly be necessary to test these ideas as explanations of the present data.

## 5-HT CS responses could be responsible for inhibiting perseverative responding

5-HT could thus contribute to cognitive flexibility not only through learning and plasticity, but also by directly suppressing activity in systems responsible for violated predictions. Indeed, 5-HT has been strongly associated with suppressing both impulsive and perseverative responses through 'behavioral inhibition' (*Clarke et al., 2007*; *Boureau and Dayan, 2011*; *Cools et al., 2011*). In addition to US signals that could explain the contribution of 5-HT to uncertainty-driven learning, we also found CS or cue responses that could explain a direct and immediate contribution to behavioral control during environmental change. We found that 5-HT CS responses, like DA CS responses, were

strongly positively correlated with CS value, consistent with previous reports (*Liu et al., 2014*; *Cohen et al., 2015*; *Hayashi et al., 2015*). Indeed, 5-HT and DA CS signals were qualitatively extremely similar, both after initial training and after relearning. Given that 5-HT and DA are thought to drive opposing processes of behavioral inhibition and invigoration, respectively (*Boureau and Dayan, 2011*; *Cools et al., 2011*), this would suggest that the two systems effectively cancel one another out. However, surprisingly, we found that the CS responses of 5-HT neurons were not only much slower than DA neurons to adapt to new associations after the reversal, but were also maintained throughout the extinction of the maladaptive perseverative response, as would be needed to prevent interference of the old appetitive response. Furthermore, there was a significant correlation across animals in the post-reversal learning rates of trial-by-trial 5-HT activity and that of anticipatory licking (*Figure 10*).

This difference in rates of adaptation between the two systems, which to our knowledge was not previously reported in any neuromodulatory system, implies that the net balance between DA and 5-HT will undergo specific dynamics during learning that resemble the classical proposal concerning opponent processes by Solomon & Corbit (1974). Specifically, because DA cue responses are quicker to establish, cues undergoing positive changes in outcome value will temporarily favor DA signals. Conversely, because DA cue responses are also quicker to withdraw, cues undergoing negative changes in outcome value will temporarily favor 5-HT signals. Thus, positive changes will favor DA and behavioral invigoration, and negative changes will favor 5-HT and behavioral suppression. This may explain why 5-HT is specifically critical in preventing responses to cues that were previously rewarding, which is observed experimentally (*Figure 1*; *Clarke et al., 2007*). The origins of the differences in 5-HT and DA learning dynamics will be important to uncover, and might arise from differences in the systems feeding into the two neuromodulators. Interestingly, neural responses in the caudate nucleus, a major recipient of DA projections (*Clarke et al., 2011*), adapt faster during reversal learning, while the PFC, a major target of 5-HT projections (*Muzerelle et al., 2016*), adapts more slowly (*Pasupathy and Miller, 2005*).

## Implications of neuronal heterogeneity and other complexities of the 5-HT system

The technique of fiber photometry of genetically-encoded calcium indicators provides excellent genetic specificity and stable long-term recordings, but does not allow the resolution of single-neuron responses. It is therefore possible that differential activity patterns within specific subpopulations of DRN 5-HT neurons exist that could not be resolved by this recording method. In fact, several studies point to a heterogeneity among DRN neurons, both in terms of physiological responses (*Ranade and Mainen, 2009*) and in terms of projection targets of DRN cell groups (*Muzerelle et al., 2016*) and single neurons (*Gagnon and Parent, 2014*). This would suggest that the different signals we observed—for example, CS vs. US or rewarding vs. aversive USs—could have different origins and functions.

Even if this is the case to some extent, and given the consistency of our optical fiber targeting (*Figure 3—figure supplement 1*), we believe that such heterogeneity probably will not substantially impact our conclusions for several reasons. First, importantly, we established that the population from which we are recording contributes to reversal learning, and it is therefore a relevant population. Second, activity patterns were consistent across mice (*Figure 3—figure supplements 3* and *5*, *Figure 4*, *Figure 4—figure supplement 1*, *Figure 5*, *Figure 5—figure supplement 1*, *Figure 9—figure supplements 1* and *2*), despite inevitable small variations in infections and fiber placements (*Figure 3—figure supplement 1*), indicating that the findings are robust to the precise population monitored. Third, single-unit recordings (*Cohen et al., 2015*; *Hayashi et al., 2015*) show that rewarding and aversive events activate the same individual DRN neurons (including identified 5-HT neurons), and are therefore not generated by distinct populations. Finally, because individual 5-HT neurons have broad projection fields (*Muzerelle et al., 2016*) and transmit primarily by volume conduction (*Dankoski and Wightman, 2013*), heterogeneity will tend to be averaged out through pooling by downstream targets.

Another limitation of our study relates to the pharmacogenetic approach to inhibiting 5-HT neurons. While it has good genetic specificity, its spatial resolution is limited by the spread of the viral particles containing the hM4Di receptor in the DRN, and its temporal resolution is on the order of dozens of minutes. Additionally, although we know this approach should inhibit 5-HT neurons in vivo

(*Teissier et al., 2015*), we did not test the efficacy of this inhibition in our animals. The limited temporal resolution of this approach makes it impossible to distinguish the contribution of CS and US 5-HT signals to behavioral flexibility. Still, we have an indication that CS responses might play a role in behavioral inhibition of perseverative responding. This could potentially be resolved in future experiments using optogenetic inhibition.

### Implications for the DA–5-HT opponency theory

Our results support, in a general sense, the long-standing notion of DA–5-HT opponency (*Boureau and Dayan, 2011*), but call for a refinement of this view. Rather than carrying the positive and negative sides of a single-signed prediction error (*Daw et al., 2002*; *Boureau and Dayan, 2011*; *Cools et al., 2011*), DA–5-HT opponency seems to be more complex and subtle. As has been classically described, the activity of DA neurons which we recorded closely resembled a so-called signed RPE (*Schultz et al., 1997*). The US-related 5-HT signals, on the other hand, resemble in important respects, but don't perfectly match, the concept of an 'unsigned RPE' signal. Thus, 5-HT neurons responded not to an opposing class of events, but to an overlapping and broader range of events compared to DA. In this respect, they might be acting as a kind of inhibitory 'surround' to DA's excitatory 'center', helping to sharpen the focus of behavioral attention. Nevertheless, just as DA signaling is increasingly acknowledged to be more complex than classically described (*Cohen et al., 2012*; *Eshel et al., 2015*; *2016*; *Matsumoto et al., 2016*; *Wise, 2004*), attributing a single function to 5-HT neurons is also clearly an oversimplification.

With respect to CS responses, 5-HT neurons showed a remarkably similar pattern of activity to that of DA neurons, scaling closely with the value of the stimuli. A possible explanation for this observation is that 5-HT CS responses could be learned by the same DA-dependent process that generates DA CS responses. If this were the entire story, then 5-HT and DA CS responses might simply balance and nullify one another. However, the fact that 5-HT CS responses evolved much more slowly than did DA CS responses means that such a balance will not hold in dynamic environments. This dynamic balance between positive and negative forces resembles the balance of excitation and inhibition in the cortex (for example, *Wehr and Zador, 2003*), albeit on a much slower time scale. Such a temporal asymmetry between opponent processes endows the joint system with novel and potentially important dynamics, which may be an important substrate in the dynamics of learning, as previously proposed (Solomon and Corbit, 1974). CS and US responses of a similar nature to those observed in 5-HT and DA neurons also appear to be observed in other neuromodulatory systems as well (*Yu and Dayan, 2005*; *Dayan and Yu, 2006*; *Sara and Bouret, 2012*; *Hangya et al., 2015*). This suggests that, contrary to the notion that each neuromodulator reports a completely distinct signal (*Daw et al., 2002*; *Doya, 2008*; *Dayan, 2012*), they have highly overlapping signals, presumably derived from partly overlapping inputs, but with more subtle differences through which their joint actions are orchestrated.

This description of the dynamics of 5-HT neurons during reversal learning provides novel insights into how this system can contribute to cognitive flexibility. Moreover, the results also suggest the need for a refinement in conventional conceptions of 5-HT's function in the regulation of mood, with implications for understanding its role in depression and other psychiatric disorders. More than reporting the affective value of the environment (*Boureau and Dayan, 2011*; *Luo et al., 2016*), we suggest that 5-HT facilitates the ability of an organism to adapt flexibly to dynamic environments through plasticity and behavioral control. The clinical benefits of an enhancement of 5-HT function would therefore stem not from directly biasing affective states toward the positive, but by preventing the negative consequences of maladaptive world views and facilitating adaptive change (*Branchi, 2011*).

## Materials and methods

### Animals

All procedures were reviewed and performed in accordance with the European Union Directive 2010/63/EU and the Champalimaud Centre for the Unknown Ethics Committee guidelines, and approved by the Portuguese Veterinary General Board (Direcção Geral de Veterinária, approvals 0420/000/000/2011 and 0421/000/000/2016). Thirty-four C57BL/6 male mice between two and nine

months of age were used in this study. Mice resulted from the backcrossing of BAC transgenic mice into Black C57BL for at least six generations, and expressed the Cre recombinase under the control of specific promoters. Twenty-six mice expressed Cre under the serotonin transporter gene (Tg (*Slc6a4*-cre)ET33Gsat/Mmucd) from GENSAT (*Gong et al., 2007*); RRID:MMRRC_017260-UCD), four mice under the tyrosine hydroxylase gene, two mice (Tg(*Th*-cre)FI12Gsat/Mmucd) from GENSAT (*Gong et al., 2007*); RRID:MMRRC_017262-UCD), and two mice (B6.Cg-Tg(*Th*-Cre)1Tmd/J) from the Jackson Laboratory (*Savitt et al., 2005*); RRID:IMSR_JAX:008601). Animals (25–45 g) were group-housed prior to surgery and individually housed post-surgery and kept under a normal 12 hr light/dark cycle. All experiments were performed in the light phase. Mice had free access to food. After training initiation, mice used in behavioral experiments had water availability restricted to the behavioral sessions.

## Stereotaxic viral injections and fiber implantation

Mice were deeply anaesthetized with isoflurane mixed with $O_2$ (4% for induction and 0.5–1% for maintenance) and placed in a stereotaxic apparatus (David Kopf Instruments). Butorphanol (0.4 mg/kg) was injected subcutaneously for analgesia and Lidocaine (2%) was injected subcutaneously before incising the scalp and exposing the skull. For SERT-Cre mice a craniotomy was drilled over lobule 4/5 of the cerebellum, and a pipette filled with a viral solution was lowered to the DRN (bregma −4.55 anteroposterior (AP), −2.85 dorsoventral (DV)) with a 32° angle toward the back of the animal. For the two TH-Cre mice from The Jackson Laboratory, the pipette was targeted to the VTA (bregma −3.3 AP, 0.35 mediolateral (ML), −4.2 DV) with a 10° lateral angle, and for the two TH-Cre mice from GENSAT we targeted the SNc (bregma −3.15 AP, 1.4 ML, −4.2 DV). Although the TH-Cre lines have been characterized as less specific than other DA-specific lines (*Lammel et al., 2015*), we targeted our fibers to areas where this specificity problem is reduced (*Lammel et al., 2015*) and that are known to contain the classical DA neurons that show RPE activity and are involved in reward processing (*Lammel et al., 2011*, *2012*; *Matsumoto and Hikosaka, 2009*; *Lerner et al., 2015*; *Kim et al., 2016*).

Viral solution was injected using a Picospritzer II (Parker Hannifin) at a rate of approximately 38 nl per minute. The expression of hM4D and of all fluorophores was Cre-dependent, and all viruses were obtained from the University of Pennsylvania (with $10^{12}$ or $10^{13}$ GC/mL titers). For hM4D experiments 1 µl AAV2/5 - Syn.DIO.hM4D.mCherry was injected in the DRN of 8 SERT-Cre mice. No virus was injected in WT controls (n = 4). For analysis of GCaMP6s specific expression in 5-HT neurons, four SERT-Cre mice were transduced in the DRN with 1 µl of viral stock solution of AAV2/1 - Syn.Flex.GCaMP6s.WPRE.SV40. For behavioral experiments in control mice (four SERT-Cre mice), 1.5 µl of a mixture of equal volumes of AAV2/1.EF1a.DIO.eYFP.WPRE.hGH and of AAV2/1.CAG. FLEX.tdTomato.WPRE.bGH was used. For the remaining mice, a mixture of equal volumes of AAV2/(1 or 9).Syn.Flex.GCaMP6s.WPRE.SV40 and of AAV2/1.CAG.FLEX.tdTomato.WPRE.bGH was injected: 1.5 µl in ten SERT-Cre mice (distributed around six points around the target coordinates) and 0.75 µl of 10 times diluted mixture in four TH-Cre mice (distributed around four points around the target).

For photometry experiments, optical fiber implantations were done after infection and a head plate for head fixation was placed above Bregma; the skull was cleaned and covered with a layer of Super Bond C and B (Morita). An optical fiber (300 µm, 0.22 NA) housed inside a connectorized implant (M3, Doric Lenses) was inserted in the brain, with the fiber tip positioned at the target for SERT-Cre mice and 200 µm above the infection target for TH-Cre mice. The implants were secured with dental acrylic (Pi-Ku-Plast HP 36, Bredent).

## Behavioral training and testing protocol

Mice were water-deprived in their home cage on the day of surgery, or up to five days before it. During water deprivation each mouse's weight was maintained above 80% of its original value. Following infection and implantation surgery, mice were habituated to the head-fixed setup by receiving water every 4 s (6 µl drops) for three days, after which training in the odor-guided task started. A mouse nose poke (007120.0002, Island Motion Corporation) using an IR photoemitter-photodetector was adapted to measure licking as IR beam breaks. To deliver air puffs, a pulse of air was delivered through a tube to the right eye of the mouse. Sounds signaling the beginning of the trial and

the outcomes were amplified (PCA1, PYLE Audio Inc.) and presented through speakers (Neo3 PDRW W/BC, Bohlender-Graebener). Water valves (LHDA1233115H, The Lee Company) were calibrated and a custom made olfactometer designed by Z.F.M. (Island Motion) was used for odor delivery. The behavioral control system (Bcontrol) was developed by Carlos Brody (Princeton University) in collaboration with Calin Culianu, Tony Zador (Cold Spring Harbor Laboratory) and Z.F.M. Odors were diluted in mineral oil (Sigma-Aldrich) at 1:10 and 25 µl of each diluted odor was placed inside a syringe filter (2.7 µm pore size, 6823–1327, GE Healthcare) to be used in two sessions (~100 trials for each odor). Odorized air was delivered at 1000 ml/min. Odors used were carvone (R)-(-), 2-octanol (S)-(+), amyl acetate and cuminaldehyde. For the behavioral task used in the hM4D experiment, these odors were associated with reward, reward, nothing and nothing, respectively. For the behavioral task used in the GCaMP6s experiment, they were associated with a large reward (4 µl water drop), small reward (2 µl water drop), neutral (no outcome) and punishment (air puff to the eye) before reversal, and with punishment, neutral, small reward and large reward after the reversal of the cue–outcome associations, respectively. In each trial, white noise was played to signal the beginning of the trial and to mask odor valve sounds. A randomly selected odor was presented for 1 s. Following a 2 s trace period, the corresponding outcome was available. Mice completed one session per day. For hM4D experiments, odors were introduced in pairs. For photometry experiments, training started by presenting only the large and small reward trials to the mice, followed by the introduction of the neutral type of trial in the next session, and finally the punishment trial in the following one. Punishment trials were presented gradually until all four types of trials had the same probability of occurrence and each session consisted of 140–346 trials (minimum to maximum, 223 ± 30, mean ± SD). Time to odor (foreperiod), trace period and inter-trial interval (ITI) were also gradually increased during training until mice could do the task with their final values: foreperiod was 3 to 4 s, taken from a uniform distribution, trace was fixed at 2 s, ITI was 4 to 8 s taken from a uniform distribution.

hM4D experiments were run in two batches: the experiment was run first on the WT animals and then on the SERT-Cre animals (with some overlapping days). Photometry experiments were run in five batches in the following sequence: 3 (SERT-Cre, experimental)+3 (SERT-Cre, experimental and YFP controls)+2 (SERT-Cre YFP controls)+6 (SERT-Cre, experimental)+4 (TH-Cre, experimental).

For the hM4D experiments, mice received a daily injection of vehicle (saline 0.9% and DMSO 0.25%) 40 min. before session start. The volume of these daily injections of vehicle was determined according to each mouse's weight, and it required an adjustment of the water drop size for each mouse to keep them motivated to do 150 trials per session. On the reversal day and the two following days, for experimental mice, CNO was diluted in the vehicle solution and delivered at a concentration of 3 mg/kg. In both reversal learning tasks used, we ensured that mice could correctly perform the task on at least three consecutive days before reversing the odor–outcome contingency for the first time. On the reversal day, mice started the session as before and the contingencies were reversed at trial 50 in the hM4D experiment, and between the 32th and the 100th trial (73 ± 12, mean ± SD) in the GCaMP6s experiments. One SERT-Cre mouse was excluded from the hM4D analysis for not showing a differential lick rate within 1.5 s of US delivery, between odors 1 and 2 (rewards) and odors 3 and 4 (nothing). Two mice were excluded from the GCaMP6s data analysis for bad fiber placement assessed after histology analysis (more than 400 µm away from the infection area): one SERT-Cre and one TH-Cre mouse. Additionally, another SERT-Cre mouse was discarded from the reversal data analysis because of experimental problems with the fiber during the reversal session. In four SERT-Cre mice and in all TH-Cre mice, at five to six days after the reversal, we introduced uncued US trials during the task. These trials represented approximately 20% of the total number of trials in a session during which no odor cue was presented; the typical white noise of the foreperiod was immediately followed by one of the four possible outcomes, randomly selected (11 ± 4 uncued vs 44 ± 8 cued trials per session, mean ± SD). To analyze these data, four sessions with cued and uncued outcomes were pooled together for each mouse. All GCaMP6s experiments were performed within the limit of one month from the viral injection date, to avoid cell death due to over-expression of GCaMP6s in neurons.

## Fiber photometry setup

The dual-color fiber photometry acquisition setup consists of a three-stage tabletop black case containing optical components (filters, dichroic mirrors, collimator), two light sources for excitation and

two photomultiplier tubes (PMTs) for acquisition of fluorescence of a green (GCaMP6s) and of a red (tdTomato) fluorophore.

We used a 473 nm (maximum power: 30 mW) and a 561 nm (maximum power: 100 mW) diode-pumped solid-state laser (both from Crystalaser) for excitation of GCaMP6s and of tdTomato, respectively. Beamsplitters (BS007, Thorlabs) and photodiodes (SM1PD1A, Thorlabs) were used to monitor the output of each laser. The laser beams were attenuated with absorptive neutral density filters (Thorlabs), and each was aligned to one of the two entrances of the three-stage tabletop black case (Doric Lenses). At the corresponding entrances the excitation filters used were 473 nm (LD01-473/10-25 Semrock) and 561 nm (LL02-561-25 Semrock). Inside the black case three interchangeable/stackable cubes (Doric Lenses) with dichroic mirrors were used: one to separate the 473 nm excitation light from longer wavelengths (Chroma T495LP), one to collect the emission light of GCaMP6s (FF552-Di02−25 × 36 Semrock), and one to separate the 561 nm excitation light from tdTomato's fluorescence (Di01-R561−25 × 36). A collimator (F = 12 mm, NA = 0.50, Doric Lenses) focused the laser beams in a single multimode silica optical fiber with 300 μm core and 0.22 NA (MFP_300/330/900–0.22_2.5m-FC_CM3, Doric Lenses), which was used for transmission of all excitation and emission wavelengths. The three-stage tabletop black case had two exits, one for each fluorophore emission, at which we placed the corresponding emission filters (Chroma ET525/50m for GCaMP6s and Semrock LP02-568RS-25 for tdTomato), and convergent lenses (F = 40 mm and F = 50 mm, Thorlabs) before the photodetectors (photomultiplier tube module H7422-02, Hamamatsu Photonics). The output signals of the PMTs were amplified by a preamplifier (C7319, Hamamatsu), acquired in a Micro1401-3 unit at 5000 Hz and visualized in Spike2 software (Cambridge Electronic Design).

Light power at the tip of the patchcord fiber was 200 μW for each wavelength (473 nm and 561 nm) for all experiments (measured before each experiment with a powermeter PM130D, Thorlabs). This patchcord fiber was attached to the fiber cannula each animal had implanted (MFC_300/330–0.22_5 mm_RM3_FLT Fibre Polymicro, polymide removed) through a titanium M3 thread receptacle.

## Data analysis

All data were analyzed in MATLAB (RRID:SCR:001622). For the behavioral experiments, lick rate was acquired at 1 KHz and smoothed using convolution with a Gaussian filter of 50 ms standard deviation. Mean anticipatory licking was calculated for each trial as the mean lick rate in the period of 500–2800 ms after odor onset, after subtracting the mean lick rate over a baseline period of −500 to 500 ms around odor onset. To evaluate the aversiveness of the air puff delivered to the mice in the photometry experiment, we used a CCD camera (Point Grey) to record the right eye of six mice during several sessions at 60 Hz. To quantify blinking in video data, we manually selected the eye area in each session and calculated the mean pixel value for that area; then, for each frame, we subtracted this value from the previous frame to obtain a measure of movement. The start and end of blinking created a sudden increase and decrease, respectively, in the difference between the mean pixel value of consecutive video frames. In the time course analysis of the licking behavior in the hM4D experiment, trials of sessions around reversal were concatenated and smoothing over three trials was performed along the trials. For each reversed odor and each mouse, the last 50 trials before reversal were fit by a constant function of the form (A+B); the first 200 trials after the reversal were fit by an exponential function of the form (A+B*exp(-t/τ)) using *fminsearch* in MATLAB. The conditions for this fitting to be done were: the last 100 trials before reversal had to be statistically different from trials 100–200 after the reversal (*t*-test), the change in licking pattern had to follow the correct trend of the reversal (increase in licking for positive reversals and decrease in licking for negative ones), and the time constant obtained had to be larger than 1. Mouse–odor pairs that did not fulfill this condition were excluded (that is, odor 4 of mice M#4 and M#5). Time constants were grouped according to the type of reversal and genotype with drug treatment, and compared using one-way ANOVA. Then, for each SERT-Cre mouse, the time constant of the reversal with the vehicle was subtracted from the reversal with CNO. The same was done for WT mice, but subtracting the time constant of reversal two from that of reversal 1 (since CNO was delivered in both). *t*-tests were used to determine whether these differences had means significantly different from zero.

Fluorescence data were downsampled to 1 kHz and smoothed using convolution with a Gaussian filter of 100 ms standard deviation. For each trial, the relative change in fluorescence, $\Delta F/F_0 = (F-F_0)/F_0$, was calculated by taking $F_0$ to be the mean fluorescence during a 1 s period before the odor

presentation for both the red and the green channels ($[\Delta F/F_0]_{GREEN}$ and $[\Delta F/F_0]_{RED}$). For each session and each mouse, the distribution of green and red values of $\Delta F/F_0$ was fitted by the sum of two Gaussians along the red channel, and the crossing point between these two Gaussians was used as a boundary (excluding the first and last 1000 ms of each trial because of filtering artifacts). All values of $[\Delta F/F_0]_{RED}$ below this boundary were used, together with the corresponding $[\Delta F/F_0]_{GREEN}$, to fit a linear regression line. Then, for each trial we corrected the green $\Delta F/F_0$ values using the parameters ($a$ - slope; $b$ - offset) obtained with the regression model of that mouse in that session: $[\Delta F/F_0]_{GREEN\_corr} = [\Delta F/F_0]_{GREEN} - a*[\Delta F/F_0]_{RED} - b$.

Behavioral data were organized as a function of US type and divided into CS and US responses. $[\Delta F/F_0]_{GREEN\_corr}$ US responses were normalized by subtracting the mean $[\Delta F/F_0]_{GREEN\_corr}$ over the 1 s interval before US onset. The CS or US response was considered the mean of the signal during the 1.5 s period after CS or US onset, respectively. For each mouse, all CS and US responses were z-scored in the expert phase, so that the amplitudes of responses to the different events could be compared. Analysis of US responses across days was performed by z-scoring all US responses of each mouse across days for each US type. Statistical analysis was done by comparing each day with pre-reversal days −1 and −2. For each mouse, mean amplitude of response to each US on the reversal day was also compared to the day before the reversal. For analysis of uncued US trials, four days of each mouse were pooled together due to the small number of uncued trials of each US type in each session.

For the analysis of CS response time courses during a reversal, each mean amplitude change across trials was fitted by an exponential function with maximum time constant of 225 trials (minimum number of trials after the reversal for any US type of any mouse). The same criteria and parameters used for the hM4D experiments were used here. Time constants for mouse–odor pairs were pooled together in pairs (odors 1 and 2, and odors 3 and 4) which correspond to the negative and positive reversals, respectively.

The data are available from the Dryad Digital Repository: 10.5061/dryad.649nk (*Matias, 2016*).

## Immunohistochemistry and anatomical verification

Mice were deeply anesthetized with pentobarbital (Eutasil, CEVA Sante Animale), exsanguinated transcardially with cold saline and perfused with 4% paraformaldehyde (P6148, Sigma-Aldrich). Coronal sections (40 µm) were cut with a vibratome and used for immunohistochemistry. For SERT-Cre mice used in expression specificity analysis, anti-5-HT (36 hr incubation with rabbit anti-5-HT antibody 1:2000, Immunostar, RRID:AB_572263, followed by 2 hr incubation with Alexa Fluor 594 goat anti-rabbit 1:1000, Life Technologies) and anti-GFP immunostaining (15 hr incubation with mouse anti-GFP antibody 1:1000, Life Technologies, followed by 2 hr incubation with Alexa Fluor 488 goat anti-mouse 1:1000, Life Technologies) were performed sequentially. For SERT-Cre mice used in behavioral experiments, anti-GFP immunostaining was performed (15 hr incubation with rabbit polyclonal anti-GFP antibody 1:1000, Life Technologies, followed by 2 hr incubation with Alexa Fluor 488 goat anti-rabbit 1:1000, Life Technologies).

For TH-Cre mice, anti-GFP (15 hr incubation with rabbit polyclonal anti-GFP antibody 1:1000, Life Technologies, followed by 2 hr incubation with Alexa Fluor 488 goat anti-rabbit 1:1000, Life Technologies) and anti-TH immunostaining (15 hr incubation with mouse monoclonal anti-TH antibody 1:5000, Immunostar, RRID:AB_572268, followed by 2 hr incubation with Alexa Fluor 647 goat anti-mouse, 1:1000, Life Technologies) were performed sequentially.

To quantify the specificity of GCaMP6s expression in 5-HT neurons of SERT-Cre mice, we used a confocal microscope (Zeiss LSM 710, Zeiss) with a 20X objective (optical slice thickness of 1.8 µm) to acquire z-stacks of three slices around the center of infection. Images for DAPI, GFP and Alexa Fluor 592 were acquired, and cells expressing GCaMP6s and cells stained with 5-HT antibody were quantified in a $200 \times 200$ µm window in the center of the DRN. The same was done for quantification of specificity in DA neurons of TH-Cre mice, but acquiring Alexa Fluor 647 instead of 592, and taking the $200 \times 200$ µm window on the infection side. To evaluate fiber location in relation to infection, images for DAPI, YFP or GFP and tdTomato were acquired with an upright fluorescence scanning microscope (Axio Imager M2, Zeiss) equipped with a digital CCD camera (AxioCam MRm, Zeiss) with a 10X objective. The location of the fiber tip was determined by the most anterior brain damage made by the optical fiber subtracted by its radius. The center of infection was estimated through visual inspection of slices as the location where there were most infected cells. The distance between

the fiber tip location and center of infection was calculated as an anterior–posterior distance, which was estimated by comparing each corresponding location in the mouse brain atlas (*Paxinos and Franklin, 2001*). To determine the overlap between cells expressing YFP or GCaMP6s and tdTomato in SERT-Cre mice, we used a confocal microscope (Zeiss LSM 710, Zeiss) with a 20X objective (optical slice thickness of 1.8 μm) to image three slices around the center of infection (slices −1, 0 and 1, relative to it). All cell counts were done using the Cell Counter plugin of Fiji (RRID:SCR_002285).

## Acknowledgements

We thank R M Costa and J J Paton Labs for TH-Cre mice, Susana Dias and Sérgio Casimiro for histology and immunohistochemistry assistance, and Dario Sarra for running behavioral experiments for a few days. We also thank C Poo, B V Atallah, M Murakami, G Agarwal and J J Paton for comments on a previous version of the manuscript, and all members of the Systems Neuroscience Lab and the Champalimaud Research for useful discussions and feedback during the development of this project. This work was supported by the Fundação para a Ciência e Tecnologia (fellowship SFRH/BD/43072/2008 to SM), Human Frontier Science Program (fellowship LT000881/2011L to EL), European Research Council (Advanced Investigator Grants 250334 and 671251 to ZFM) and Champalimaud Foundation (ZFM).

## Additional information

### Funding

| Funder | Grant reference number | Author |
|---|---|---|
| Fundação para a Ciência e a Tecnologia | SFRH/BD/43072/2008 | Sara Matias |
| Human Frontier Science Program | LT00088/011L | Eran Lottem |
| European Research Council | 250334 | Zachary F Mainen |
| Champalimaud Foundation | | Zachary F Mainen |
| European Research Council | 671251 | Zachary F Mainen |

The funders had no role in study design, data collection and interpretation, or the decision to submit the work for publication.

### Author contributions

SM, Conceptualization, Data curation, Formal analysis, Validation, Investigation, Visualization, Methodology, Writing—original draft, Writing—review and editing; EL, Conceptualization, Data curation, Formal analysis, Supervision, Validation, Investigation, Visualization, Methodology, Writing—review and editing; GPD, Supervision, Methodology; ZFM, Conceptualization, Resources, Supervision, Funding acquisition, Visualization, Writing—original draft, Project administration, Writing—review and editing

### Author ORCIDs

Sara Matias, http://orcid.org/0000-0002-7432-6754
Eran Lottem, http://orcid.org/0000-0001-5852-928X
Guillaume P Dugué, http://orcid.org/0000-0002-4106-6132
Zachary F Mainen, http://orcid.org/0000-0001-7913-9109

### Ethics

Animal experimentation: This study was performed in strict accordance with the European Union Directive 2010/63/EU. All animals were handled according to approved institutional animal care and use guidelines by the Champalimaud Centre for the Unknown Ethics Committee. The protocol was approved by the Portuguese Veterinary General Board (Direccao Geral de Veterinaria, approvals 0420/000/000/2011 and 0421/000/000/2016).

## Additional files

### Major datasets

The following dataset was generated:

| Author(s) | Year | Dataset title | Dataset URL | Database, license, and accessibility information |
|---|---|---|---|---|
| Sara Matias, Eran Lottem, Guillaume P Dugué, Zachary F Mainen | 2016 | Data from: Activity patterns of serotonin neurons underlying cognitive flexibility | http://dx.doi.org/10.5061/dryad.649nk | Available at Dryad Digital Repository under a CC0 Public Domain Dedication |

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
