## [Decision Letter]

Thank you for submitting your article "Firing patterns of serotonin neurons underlying cognitive flexibility" for consideration by *eLife*. Your article has been favorably evaluated by Timothy Behrens (Senior Editor) and three reviewers, one of whom is a member of our Board of Reviewing Editors. The following individual involved in review of your submission has agreed to reveal his identity: Jeremiah Y Cohen (Reviewer #2).

The reviewers have discussed the reviews with one another and the Reviewing Editor has drafted this decision to help you prepare a revised submission.

Summary:

This study by Mainen and colleagues examines the role of 5HT in reversal learning. They use a combination of techniques, including inactivation, imaging, and behavior in rodents, to show that dorsal raphe 5HT neurons encode both positive and negative prediction errors and may play a role in modulating perseverative behaviors. A particularly noteworthy feature of this study is that they compared response properties of 5HT and DA neurons, showing that they share many similarities but some key differences that help to distinguish their specific computational roles in adaptive behavior.

The reviewers agreed that the experiments are well done, the analyses sound, and the paper well written. The reviewers also commended the novel idea of how different time courses of learning could cause 5-HT to regulate behavior during certain forms of reversal.

Essential revisions:

1) The claim that "5-HT US responses resemble closely an unsigned prediction error" seems like an over-simplification, given the data. Several complications to this interpretation should be addressed more directly, including:

The very next paragraph in the Discussion notes the "one notable divergence" from this idea, involving the response to the air puff. The argument that this difference involves a difference in control is interesting, but not tested directly. Also, what about blinks, which presumably could reduce the aversiveness of the air puffs. And what is the evidence that rodents learned to predict the air puffs as well as the positive rewards? Moreover, consider the reversals between the large reward and air puff. In theory, both unexpected outcomes should generate equal unsigned prediction errors. However, only the response to the large reward is higher when unpredicted. Response to the air puff is unchanged, with a trend to being lower. Even if animals failed to respond to air puffs appropriately (as the authors suggest in the Discussion), they should still detect that on unexpected air puff trials there is an omission of big reward. We know they have detected this omission and learned from it because they rapidly reduced licking to the CS. This should entail a large unsigned prediction error to omission of big reward at least as large as the clear response to omission of small reward. Indeed, DA had similar inhibitory responses to unexpected neutral and air puff USs. However 5-HT's air puff expectation effect was non-significant and the response actually trended to be larger for expected air puff, which is the opposite effect from what it should have. It seems like the most parsimonious account is that 5-HT reports unsigned prediction error except when an air puff is present, in which case it ignores the unsigned prediction error and has a stereotyped excitation. This would be a very interesting wrinkle on the conclusions, because the authors argue throughout the paper that 5-HT is important for learning, so there should be a big difference in reward learning when air puff is present versus absent. However, the present experiments don't provide evidence of this.

The statement that "5-HT neurons showed little or no response to expected water rewards before reversal, but responded robustly to the same rewards when they were unexpected, after reversal" seems problematic. Specifically, unexpected small reward did not seem to have a robust response that was affected by expectation (Figure 2, Figure 3—figure supplement 1). If 5-HT reported unsigned prediction error this response should be significant as it was in DA neurons.

In Figure 3, Figure 5-HT was activated by unexpected neutral tone but DA did not respond (at least as indicated in Figure 3 (Figure 3—figure supplement 1 seems to show mild inhibition for unexpected neutral and air puff, so the absence of effect in Figure 3 may be an issue with the analysis windows). According to the logic of the paper, either 5-HT does not signal unsigned prediction error, or it is based on a different concept of prediction error than DA.

If 5-HT reports unsigned prediction error, "surprise", or "uncertainty", why is it activated by CSs in proportion to their value? If it reported unsigned prediction error then it should be more activated by CSs with extreme values than ones with intermediate values because those evoke the biggest unsigned prediction errors. If it reported surprise then it should be excited similarly by all CSs because all four CSs are equally probable. If it reported uncertainty then it should be more activated by all CSs shortly after reversal when their CS-US associations became more uncertain.

2) The statement that "rather than reporting the affective value of the environment…, we suggest that 5-HT facilitates the ability of an organism to flexibly adapt to dynamic environments through plasticity and behavioural control" is a very intriguing suggestion made throughout the paper. However, it is not tested directly, but perhaps could be with the existing data. Were correlations between 5-HT activity and behavioral flexibility? 5-HT responds more to positive than negative CSs but animals seem to learn about both with similar speed (at least in Figure 4). 5-HT responds more to unexpected neutral than unexpected small reward, and more to unexpected large reward than unexpected air puff, so if the authors were correct animals should learn at different rates from those outcomes. This could be tested. Also, the authors measure learning rates for CS but not US responses. Are their learning rates consistent? This seems highly relevant to their interpretations. For instance 5-HT CS responses are slow to learn but Figure 2 makes US learning seem very fast. Does this mean that the US "surprise" response and CS "value" response are based on different expectations? If so this would be further evidence that the 5-HT responses obey different rules, and may not be explained by a single umbrella concept like "surprise" or "unsigned prediction error".

3) Given that we don't know the underlying distribution of single-5-HT-neuron calcium dynamics, it is difficult to interpret the point estimate of that distribution (population calcium dynamics measured here). This is especially salient in the context of electrophysiological recordings of individual DR neurons (including work from the senior author's lab [Ranade and Mainen, 2009]), that show a high degree of heterogeneity of individual DR (and identified 5-HT) neuronal responses. This is treated in the Discussion, but the limitations need to be more clearly defined. Indeed, Muzerelle et al., cited here to support the idea of diffuse projection targets of individual 5-HT neurons, actually shows quite a bit of target specificity, and agrees with work from Gagnon and Parent (PLoS ONE, 2014). Also, I strongly suggest changing the title, as firing patterns were not measured in this study.

4) The argument that previous studies did not find prediction-error-like responses due to weak prediction errors could be more clearly explained. Why is reversal a "stronger" prediction error than unexpected rewards or omissions? If anything, it seems that reversals as presented here are relatively weak in driving behavioral changes, because it took mice a couple of hundred trials to achieve asymptotic behavior. Could the different timescales of reversals compared to momentary deviations from predicted outcomes be the more relevant discrepancy? How is prediction error strength quantified here?

5) In Figure 1, it appears that there may be a small difference in the time constant fit to reversals from negative to positive. Is this true for particular mice? Specifically, were any of the effects of inactivation on the time constant of positive reversal significant? Showing mouse-by-mouse data would be useful.

6) The authors should consider rearranging their figures for publication in *eLife*, which allows more than 4 figures. The figures are all very dense and are associated with up to 9(!) supplemental figures. Also, the critical comparison between 5-HT and DA is hard to understand because no DA CS or US traces are shown in the main figures. At the very least, the authors should present a direct comparison of the traces of 5-HT and DA responses before, during, and after reversal, which is the key result of the paper.

[Editors' note: further revisions were requested prior to acceptance, as described below.]

Thank you for resubmitting your work entitled "Activity patterns of serotonin neurons underlying cognitive flexibility" for further consideration at *eLife*. Your revised article has been favorably evaluated by Timothy Behrens (Senior Editor) and a Reviewing Editor.

The manuscript has been improved but there are some remaining issues that need to be addressed before acceptance, as outlined below:

1. Figure 1—figure supplement 1: presumably the y labels should be "Anticp. licking (norm. fit)", not "Time constant (trials)," correct?

2) Figure 2: how is "eye movement or blinking" measured? What are the units that allowed those two measures to be combined (and then why is it listed as "a.u." in the figure)?

3) Figure 3—figure supplement 2: panels A and B are a bit hard to parse and could use some more explanation; e.g., these are (presumably) ordered by duration of the lick bout; they (presumably) show a fixed time before the onset of, and after the offset of, the lick bout; etc. Why do the trials at the bottom not appear to have "bouts"?

4) Figure 10: This analysis is an interesting and welcome addition, but perhaps a bit more and nuanced interpretation of these results would be useful. There seems to be an order of magnitude (at least) difference in how much longer the 5HT response versus the behavior persisted for a negative reversal.

---

## [Author Response]

*Essential revisions:*

*1) The claim that "5-HT US responses resemble closely an unsigned prediction error" seems like an over-simplification, given the data.*

We agree with the reviewers that we had presented a simplified interpretation. We have tried to rectify this in the revised version.

A) We all-but-eliminated mention of “unsigned RPE” from the Results, keeping more strictly to the observations and retaining just two references to RPE (subsection “DRN 5-HT neurons respond to both positive and negative US prediction errors”).

B) We included an extended explicit Discussion (subsection”Response to aversive events by DRN 5- HT neurons”) emphasizing the possibility that rewarding and aversive responses reflect two distinct signals (as proposed by the reviewers) as well as the possibility that the air puff response reflects a kind of “control error”.

C) We acknowledge explicitly when mentioning unsigned RPE in the Discussion that the data only partly resembled this (subsection “DRN 5-HT neurons are activated by both positive and negative reward prediction errors”, first paragraph and subsection “Implications for the DA–5-HT opponency theory”, first paragraph).

*Several complications to this interpretation should be addressed more directly, including:*

*The very next paragraph in the Discussion notes the "one notable divergence" from this idea, involving the response to the air puff. The argument that this difference involves a difference in control is interesting, but not tested directly.*

We agree that the main point of divergence between the US data and the idealized prediction error signal is with respect to responses to aversive events. We have reduced the strength of this claim throughout the manuscript. We now directly consider the proposal that US responses to rewarding and aversive stimuli may reflect two independent sources of information as well as the “control” hypothesis in the Discussion subsection “Response to aversive events by DRN 5- HT neurons”.

*Also, what about blinks, which presumably could reduce the aversiveness of the air puffs. And what is the evidence that rodents learned to predict the air puffs as well as the positive rewards?*

Mice did not show anticipatory blinking to air puffs, despite extensive training. They did show air puff triggered blinking that was present before training. Therefore, one potential explanation for the continued presence of air puff responses after training and the relative insensitivity of the air puff response to reversals is that the air puff US was never predictable. Indeed, there is evidence other previous studies that with a 2 s trace period mice do not learn predictive eye blink responses (Reynolds 1945; Boneau 1958; Cohen et al., 2012, 2015). However, it does seem puzzling that mice would learn CS-US relationships with the same trace period for rewards but not for air puffs. We now discuss this issue explicitly in the Discussion (subsection “Response to aversive events by DRN 5-HT neurons”, first paragraph).

*Moreover, consider the reversals between the large reward and air puff. In theory, both unexpected outcomes should generate equal unsigned prediction errors. However, only the response to the large reward is higher when unpredicted. Response to the air puff is unchanged, with a trend to being lower. Even if animals failed to respond to air puffs appropriately (as the authors suggest in the Discussion), they should still detect that on unexpected air puff trials there is an omission of big reward. We know they have detected this omission and learned from it because they rapidly reduced licking to the CS. This should entail a large unsigned prediction error to omission of big reward at least as large as the clear response to omission of small reward. Indeed, DA had similar inhibitory responses to unexpected neutral and air puff USs. However 5-HT's air puff expectation effect was non-significant and the response actually trended to be larger for expected air puff, which is the opposite effect from what it should have. It seems like the most parsimonious account is that 5-HT reports unsigned prediction error except when an air puff is present, in which case it ignores the unsigned prediction error and has a stereotyped excitation.*

The reviewers raise an interesting point. After the reversal from large reward to air puff the air puff US represents not only an aversive stimulus, but also the omission of an expected reward. Even if the air puff is equally unexpected before and after reversal, the presence of the additional reward omission should generate a positive US response, just as the omission of the small US response did.

We admit that we do not have a full explanation for this phenomenon. We agree that one possibility is that the air puff has an inhibitory modulatory effect on the reward omission response. This is reminiscent of the interactions recently reported by Matsumoto et al. (2016). Another possibility is simply that the air puff response alone is already at a level close to saturation and the absence of significant additional effect reflects a ceiling effect. We now discuss these issues in the Discussion (subsection “Response to aversive events by DRN 5-HT neurons”).

*This would be a very interesting wrinkle on the conclusions, because the authors argue throughout the paper that 5-HT is important for learning, so there should be a big difference in reward learning when air puff is present versus absent. However, the present experiments don't provide evidence of this.*

We do not wish to claim and are careful not to state that the 5-HT system is solely responsible for reversal learning (see point 2 below). Therefore, the idea that the speed of learning of specific CS-US associations will depend directly on the magnitude of 5-HT US responses is not a strong implication of our interpretation of the data. We expect other systems to contribute to reversal learning and their contributions may dominate or warp these relationships. It will certainly be interesting but is beyond the scope of this paper to examine directly the difference between reversal learning with and without aversive stimuli. We do note that the time constants in the DREADD experiments, in which no air puffs were used, are a bit slower than those in the GCaMP experiments.

*The statement that "5-HT neurons showed little or no response to expected water rewards before reversal, but responded robustly to the same rewards when they were unexpected, after reversal" seems problematic. Specifically, unexpected small reward did not seem to have a robust response that was affected by expectation (Figure 2, Figure 3—figure supplement 1). If 5-HT reported unsigned prediction error this response should be significant as it was in DA neurons.*

The reviewers are correct that the response was not significant when comparing only days -1 and day 0 (previous Figure 2, now Figure 6). However, when running an ANOVA across days (previous Figure 2—figure supplement 7, now Figure 4—figure supplement 1) comparing each day post reversal with the 2 days before reversal there is a significant increase in the reversal day for the population. We substituted the example mouse in Figure 4—figure supplement 1 to make this point clearer.

Nevertheless, the size of the small reward error response was probably diminished by the failure of the small water responses to disappear even after training. We added a note to this effect in the Results section: “The response to small reward was also modulated by reward expectation (Figure 4—figure supplement 1), although to a lesser degree, perhaps due to the presence of a small response even after extensive training (Figure 3—figure supplement 3).”

*In Figure 3, Figure 5-HT was activated by unexpected neutral tone but DA did not respond (at least as indicated in Figure 3 (Figure 3—figure supplement 1 seems to show mild inhibition for unexpected neutral and air puff, so the absence of effect in Figure 3 may be an issue with the analysis windows). According to the logic of the paper, either 5-HT does not signal unsigned prediction error, or it is based on a different concept of prediction error than DA.*

As described above, we have reduced the strength of our conclusions regarding the idea that 5- HT signals a pure unsigned reward prediction error. We agree with the reviewer that the absence of significant inhibition in DA neurons to neutral and air puff US’s in this experiment (previously Figure 3; now Figure 7), is likely due to the analysis window used to calculate the response. Because the detailed analysis and interpretation of DA responses to neutral and aversive stimuli is not the focus of our manuscript and is a topic of debate among DA experts (Matsumoto & Hikosaka 2009, Lammel et al. 2012, Fiorillo 2013; Lerner et al. 2015; Kim et al. 2016, Matsumoto et al., 2016), we chose to keep the analysis window constant throughout the paper/analysis (1.5 s from outcome onset). We acknowledge that further work will be important to address these issues.

*If 5-HT reports unsigned prediction error, "surprise", or "uncertainty", why is it activated by CSs in proportion to their value? If it reported unsigned prediction error then it should be more activated by CSs with extreme values than ones with intermediate values because those evoke the biggest unsigned prediction errors. If it reported surprise then it should be excited similarly by all CSs because all four CSs are equally probable. If it reported uncertainty then it should be more activated by all CSs shortly after reversal when their CS-US associations became more uncertain.*

We agree with the reviewers that the pattern of CS and US responses does not neatly fit into a highly specific pattern of “surprise”. We have made every effort to be explicit about this in our revised Results and Discussion, as discussed above.

Still, we do not believe the distinction between the terms “unsigned prediction error”, “surprise”, and “uncertainty” is as clear in the literature as proposed in the reviewers’ comment. Therefore, we do not fully understand the specific predictions being suggested.

For example, according to Courville et al., 2006, “surprise” is a term used in associative learning approaches to name non-predicted reinforcers. According to these authors, surprise signals “change”, which leads to uncertainty about one’s model of the world in a Bayesian approach. In such an approach, surprising events that are non-reinforcers can also increase uncertainty and thus increase learning. Thus, “prediction error”, “surprise” and “uncertainty” are used more or less interchangeably in previous work.

In order to try to better avoid possible confusion, we decided to avoid as much as possible using the words “surprise” and “uncertainty” except where we discuss explicitly the meaning and relationship between these terms (Discussion).

*2) The statement that "rather than reporting the affective value of the environment…, we suggest that 5-HT facilitates the ability of an organism to flexibly adapt to dynamic environments through plasticity and behavioural control" is a very intriguing suggestion made throughout the paper. However, it is not tested directly, but perhaps could be with the existing data.*

We believe that the DREADD inhibition experiment (Figure 1) does provide direct evidence for the role for DRN 5-HT in flexible adaptation of this sort, adding to the previous work cited (Clarke 2004; Clarke et al. 2007; Boulougouris & Robbins 2010; Bari et al. 2010; Brigman et al. 2010; Berg et al. 2014).

*Were correlations between 5-HT activity and behavioral flexibility? 5-HT responds more to positive than negative CSs but animals seem to learn about both with similar speed (at least in Figure 4). 5-HT responds more to unexpected neutral than unexpected small reward, and more to unexpected large reward than unexpected air puff, so if the authors were correct animals should learn at different rates from those outcomes. This could be tested.*

We thank the reviewers for the suggestion of examining more carefully the relationship between the time constants of 5-HT and behavioral changes.

We never intended to claim that 5-HT is the *only* mechanism underlying reversal learning – we use the phrase “contributes to” cognitive flexibility deliberately for that reason (subsection “5-HT CS responses could be responsible for inhibiting perseverative responding”, first paragraph, subsection “5-HT CS responses could be responsible for inhibiting perseverative responding”, last paragraph). Therefore, we don’t expect that behavioral learning rates will have a 1:1 correspondence with the magnitude of 5-HT signals, since other signals (e.g. other neuromodulators) whose time constants are unknown are likely also contributing. Moreover, for these analyses, to improve signal-to-noise we needed to group positive and negative odour data (i.e. CS1 & 2 together and CS3 & 4 together), which prevents the suggested single-CS analysis.

However, we were motivated by this suggestion to explore other potential correlations in the data, such as the relationship between the time constant of behavioral adaptation and the 5-HT CS and US responses before or after reversal.

In the process of these analyses we found a small error in the previous analysis (Figure 4, new Figure 9), namely that we were not using the background-subtracted licking rate. After reversal some mice show an overall increase in lick rate along the entire trial, including in the period before they smell the odour. Subtracting the period just before the odour onset therefore improves the detection of odor-specific differences in anticipatory lick rate. It is then easier to see the faster adaptation to the negative reversals (stopping anticipatory licking) than to the positive reversals (starting anticipatory licking).

We found two interesting results. First, we found a significant correlation between the time constant of behavior for stopping anticipatory licking and the corresponding time constant for the 5-HT CS signal during a negative reversal. There was no such correlation during positive reversals. This supports the result of the DREADD experiment and suggests that 5-HT seems to act at least partly through its CS-mediated effects. Second, for all animals the time constant of change in 5-HT CS activity for negative reversals is slower than change in anticipatory licking, as would be needed for 5-HT to continue to suppress this response by direct behavioral inhibition.

We believe these analyses provide important new support for the involvement of 5-HT in the process of behavioral adaptation. We have included them as new Figure 10 and Results (last paragraph) and Discussion (subsection “5-HT CS responses could be responsible for inhibiting perseverative responding”, first paragraph). We again thank the reviewers for suggesting this approach.

*Also, the authors measure learning rates for CS but not US responses. Are their learning rates consistent? This seems highly relevant to their interpretations. For instance 5-HT CS responses are slow to learn but Figure 2 makes US learning seem very fast. Does this mean that the US "surprise" response and CS "value" response are based on different expectations? If so this would be further evidence that the 5-HT responses obey different rules, and may not be explained by a single umbrella concept like "surprise" or "unsigned prediction error".*

We agree with the reviewers that analysis of the time course of US responses would be very interesting. Unfortunately, the US responses are somewhat smaller and less consistent than the CS responses. For example, US responses tend to decrease over a session and then increase again in the beginning of the next session. Hence, although we attempted to, we were not able to get reasonable fits to the time course of the US’s. We now mention this point in the Results (subsection “CS responses of 5-HT neurons have slower kinetics after reversal than DA’s”, third paragraph).

*3) Given that we don't know the underlying distribution of single-5-HT-neuron calcium dynamics, it is difficult to interpret the point estimate of that distribution (population calcium dynamics measured here). This is especially salient in the context of electrophysiological recordings of individual DR neurons (including work from the senior author's lab [Ranade and Mainen, 2009]), that show a high degree of heterogeneity of individual DR (and identified 5-HT) neuronal responses. This is treated in the Discussion, but the limitations need to be more clearly defined. Indeed, Muzerelle et al., cited here to support the idea of diffuse projection targets of individual 5-HT neurons, actually shows quite a bit of target specificity, and agrees with work from Gagnon and Parent (PLoS ONE, 2014). Also, I strongly suggest changing the title, as firing patterns were not measured in this study.*

We agree with the suggestion of the reviewers to change the title of the manuscript to avoid referring to firing patterns. The new title is: “Activity patterns of serotonin neurons underlying cognitive flexibility”.

In addition, we fleshed out this section of the Discussion, “Implications of neuronal heterogeneity and other complexities of the 5-HT system”. We now more explicitly discuss the limitations of fiber photometry with respect to possible heterogeneity. We now cite Muzerelle et al. 2014 and Gagnon & Parent (2014) for the specificity of projections.

*4) The argument that previous studies did not find prediction-error-like responses due to weak prediction errors could be more clearly explained. Why is reversal a "stronger" prediction error than unexpected rewards or omissions? If anything, it seems that reversals as presented here are relatively weak in driving behavioral changes, because it took mice a couple of hundred trials to achieve asymptotic behavior. Could the different timescales of reversals compared to momentary deviations from predicted outcomes be the more relevant discrepancy? How is prediction error strength quantified here?*

We agree with the reviewer that this point could have been more clear in our manuscript. We agree that the discrepancy in timescales is a very relevant point. This issue has been previously discussed in terms of expected and unexpected uncertainty by, e.g. Yu and Dayan (2002; 2005). In omission trials, which are randomly interleaved and averaged over hundreds of trials, omissions become “expected uncertainty” or variability in the outcome. In contrast, in a reversal task, the sudden change that occurs after many days of stable conditions constitutes “unexpected uncertainty” and indeed leads to a change in behaviour. We now include a similar discussion in the last paragraph of the Introduction).

*5) In Figure 1, it appears that there may be a small difference in the time constant fit to reversals from negative to positive. Is this true for particular mice? Specifically, were any of the effects of inactivation on the time constant of positive reversal significant? Showing mouse-by-mouse data would be useful.*

We agree that it appears that there is a small difference here, but it was not statistically significant. We have added the normalized fittings for individual mice-odour pairs in Figure 1—figure supplement 1. It can be seen from those plots that in the positive reversal there is a lot of variability in the three groups of mice. Because we have only one fit for each odour of each animal, we need several mice to perform statistics. This is what we did in the manuscript and we could not reach significance (as described in Figure 1 legend): 1-way ANOVA, F2,16 = 0.34, *p* = 0.715 for positive reversal.

*6) The authors should consider rearranging their figures for publication in eLife, which allows more than 4 figures. The figures are all very dense and are associated with up to 9(!) supplemental figures. Also, the critical comparison between 5-HT and DA is hard to understand because no DA CS or US traces are shown in the main figures. At the very least, the authors should present a direct comparison of the traces of 5-HT and DA responses before, during, and after reversal, which is the key result of the paper.*

We took this suggestion and reworked both the figures and associated text. The figures have been rearranged and DA traces included in the main figures for comparison with 5-HT:

Figure 1 was not changed;

Figure 2 now shows only the behavior of the animals to the reversal task used for fiber photometry;

Figure 3 shows the experimental approach and pre-reversal data for 5-HT and DA;

Figure 4 shows 5-HT and DA US responses to large reward during reversal;

Figure 5 shows 5-HT and DA US responses to neutral outcome during reversal;

Figure 6 summarizes the comparison of mean US responses to the four outcomes on the reversal day with that on the day before reversal;

Figure 7 shows the mean US responses of 5-HT and DA to surprising and predicted outcomes;

Figure 8 compares mean CS responses before reversal and 3 days after it for both 5-HT and DA;

Figure 9 shows the time course analysis of the change in CS responses for 5- HT and DA as well as our proposed model of DA-5-HT interaction during reversal;

Figure 10 shows the relationship between time constant of adaptation of 5-HT CS responses versus the time constants of behavioral adaptation.

[Editors' note: further revisions were requested prior to acceptance, as described below.]

*The manuscript has been improved but there are some remaining issues that need to be addressed before acceptance, as outlined below:*

*1. Figure 1—figure supplement 1: presumably the y labels should be "Anticp. licking (norm. fit)", not "Time constant (trials)," correct?*

Yes, thank you for pointing it out. We have corrected the *y* label in the figure.

*2) Figure 2: how is "eye movement or blinking" measured? What are the units that allowed those two measures to be combined (and then why is it listed as "a.u." in the figure)?*

We have now restricted ourselves to the use of “blinking” in the figure and in the text. The way we measured it is described in the Materials and methods section and we expanded this description in this version of the manuscript: “To quantify blinking in video data, we manually selected the eye area in each session and calculated the mean pixel value for that area; then, for each frame, we subtracted this value from the previous frame to obtain a measure of movement. The start and end of blinking created a sudden increase and decrease, respectively, in the difference between the mean pixel value of consecutive video frames.”

Because this difference between consecutive frames is the mean pixel value of the eye area, we quantify it in arbitrary units (a.u.) in the figure.

*3) Figure 3—figure supplement 2: panels A and B are a bit hard to parse and could use some more explanation; e.g., these are (presumably) ordered by duration of the lick bout; they (presumably) show a fixed time before the onset of, and after the offset of, the lick bout; etc. Why do the trials at the bottom not appear to have "bouts"?*

Thank you for pointing out important clarifications needed in the figure legend. We have included an additional panel as an inset of Figure 3—figure supplement 2 A to show the distribution of inter-lick intervals, which is used to define the bouts. The figure legend has been made more explicit: “(A), (B) Surface plots showing raw GCaMP6s (A) and tdTomato (B) fluorescence signals aligned on the onsets of lick bouts during an example session of a SERT-Cre mouse infected with the corresponding fluorophore. Gray dots represent single licks. Inset in (A, right) shows the distribution of inter-lick intervals for this session: lick bouts were defined as sequences of licks separated by no more than 315 ms. The surface plots are aligned from longer lick bouts at the top to shorter ones at the bottom (the last ones are single licks that do not belong to any bout). Fluorescence data is shown from 2 s before to 2 s after the duration of the bouts.”

*4) Figure 10: This analysis is an interesting and welcome addition, but perhaps a bit more and nuanced interpretation of these results would be useful. There seems to be an order of magnitude (at least) difference in how much longer the 5HT response versus the behavior persisted for a negative reversal.*

Thank you for this comment. We have now added the following paragraph at the end of the Results section to provide further interpretation:

“We note that while we expected that the adaptation of 5-HT CS responses to the reversal should be at least as slow as that of anticipatory licking for the two to be causally related, the fact that it was much slower (around 8 times as slow) requires explanation. […] Alternatively, it may be the case that 5-HT CS responses could serve more than a mere motor suppression function during reversal learning, and contribute to the longer-lasting learning processes required for reversal learning (He et al. 2015), such as those that prevent spontaneous recovery following extinction training (Karpova et al., 2011).”